



# More modest peak temperatures during the Last Interglacial for both Greenland and Antarctica suggested by multi-model isotope simulations

Louise C. Sime[1], Rahul Sivankutty[1], Irene Malmierca-Vallet[1], Sentia Goursaud Oger[2,1], Allegra N. LeGrande[3], Erin L. McClymont[4], Agatha de Boer[5], Alexandre Cauquoin[6], and Martin Werner[7]

[1]Ice Dynamics and Paleoclimate, British Antarctic Survey, Cambridge, United Kingdom
[2]CEA, DAM, DIF, F-91297 Arpajon, France
[3]NASA Goddard Institute for Space Studies, New York, USA
[4]Department of Geography, Durham University, Durham, UK
[5]Department of Geological Sciences, Stockholm University, Stockholm, Sweden
[6]Institute of Industrial Science, The University of Tokyo, Kashiwa, Japan
[7]Alfred Wegener Institute, Helmholtz Centre for Polar and Marine Research, Bremerhaven, Germany

**Correspondence:** Louise C. Sime (lsim@bas.ac.uk)

**Abstract.** The Last Interglacial (LIG) period, approximately 130,000 to 115,000 years ago, represents one of the warmest intervals in the past 800,000 years. Here we simulate water isotopes in precipitation in Antarctica and the Arctic during the LIG, using three isotope-enabled atmosphere-ocean coupled climate models: HadCM3, MPI-ESM-wiso, and GISS-E2.1. These models were run following the Paleoclimate Modelling Intercomparison Project, phase 4 (PMIP4) protocol for the LIG

at 127ka (kilo-years ago), supplemented by a 3000-year Heinrich Stadial 11 (H11) experiment run with HadCM3. The long H11 simulation has meltwater from the Northern Hemisphere applied to the North Atlantic which causes large-scale changes in ocean circulation including cooling in the North Atlantic and Arctic and warming in the Southern and Global Ocean. We find that the standard 127ka simulations do not capture the observed Antarctic warming and sea ice reduction in the Southern Ocean and Antarctic regions, but they capture around half of the warming in the Arctic. The H11 simulations align better with

observations: they capture more than 80% of the warming, sea ice loss, and $\delta^{18}$O changes for both Greenland and Antarctica. Decomposition of seasonal $\delta^{18}$O drivers highlights the dominant role of sea-ice retreat and associated changes in precipitation seasonality in influencing isotopic values in all simulations, alongside a small common response to orbital forcing. We use the H11 and multi-model 127k simulations together to infer LIG surface air temperature (SAT) changes based on ice core measurements. The peak inferred LIG Greenland SAT increase is +2.89 ± 1.32 K at the NEEM ice core site. This is less than

half the previously inferred warming. Peak inferred LIG Antarctic SAT increases are +4.39 ± 1.45 K at EDC, dropping to +1.67 ± 3.67 K at TALDICE. These calculated warming values are from climate effects alone, and do not take account of any ice flow or site elevation related impacts. Coastal sites in Greenland and Antarctica appear to have experienced less warming compared with higher central regions.





## 1 Introduction

The Last Interglacial (LIG) period, or Eemian, occurring approximately 130,000 to 115,000 years ago, stands out as one of the warmest interglacials in paleoclimate records over the past 800 thousand years (Past Interglacials Working Group of PAGES, 2016). The globe experienced temperatures higher than preindustrial levels during peak LIG conditions (Hoffman et al., 2017; Fischer et al., 2018). Estimates of global mean maximum surface air temperature LIG warming of 0.8±0.5K above pre-industrial levels are supported by Hoffman et al. (2017), Fischer et al. (2018), and Shackleton et al. (2020). The magnitude of warming was greater in polar and subpolar regions compared to the global-mean value (CAPE-Last Interglacial Project Members, 2006; Sime et al., 2009; Capron et al., 2014, 2017). Sime et al. (2023) estimated the Arctic wide summer surface air temperature warming at 127 ka to be 3.7± 1.5K, when the Arctic may have been sea ice free during summer (Guarino et al., 2020; Vermassen et al., 2023); whilst Greenland surface temperatures as high as 8 ± 4 K warmer than the PI has been posited by NEEM community members (2013). Similarly, peak LIG annual mean warming over central Antarctic ice core sites of +3-6 K have been suggested (*e.g.* Jouzel et al., 2007; Sime et al., 2009; Otto-Bliesner et al., 2013; Bakker et al., 2014; Capron et al., 2014).

Alongside the increases in polar surface air temperatures, sea ice cover was reduced and sea surface temperature warmed across the Arctic and Southern Oceans at 127-128 ka compared to the preindustrial period (Holloway et al., 2017; Chadwick et al., 2023). Recently Vermassen et al. (2023) used open water species, found in Arctic marine cores, to suggest that the Arctic was likely seasonally sea ice-free during the LIG, though Arctic age models are currently characterized by significant uncertainty (Razmjooei et al., 2023). This supports the idea that sea ice feedbacks were important in amplifying the Arctic LIG warming (Guarino et al., 2020; Diamond et al., 2021, 2023). In the Southern Ocean work on diatom fossil assemblages in marine sediment cores and sea-salt sodium in Antarctic ice cores helps underpin our understanding of Antarctic sea ice (Wolff et al., 2006; Crosta et al., 2022; Chadwick et al., 2020, 2021, 2022). Gao et al. (2024a) examined four recent Southern Ocean core syntheses (from: Capron et al., 2017; Hoffman et al., 2017; Chandler and Langebroek, 2021; Chadwick et al., 2021), and estimated a Southern Ocean annual SST anomaly of around +1.3K at 127 ka.

Whilst the Arctic summer warmth and loss of sea ice during the LIG can be attributed mainly to orbital forcing (Guarino et al., 2020; Diamond et al., 2023; Sime et al., 2023; Vermassen et al., 2023), the reduced sea ice and warmer Southern Ocean is not attributable to either orbital or greenhouse gas forcing, as austral summer solar insolation was lower, and greenhouse gas concentrations were below preindustrial levels at 127 ka (Berger and Loutre, 1991; Otto-Bliesner et al., 2017). Although the Antarctic ice sheet volume and configuration remain poorly constrained (Golledge et al., 2021), modelling studies nevertheless show that a smaller than preindustial ice sheet alone is very unlikely to account for the estimated LIG sea ice loss and Southern Ocean warming (Holloway et al., 2016; Hutchinson et al., 2024). Alongside this, recent simulations run according to standard Paleoclimate Modelling Intercomparison Project (PMIP) LIG protocol (Otto-Bliesner et al., 2017), which applies 127 ka orbital and greenhouse gas forcing, also do not fully replicate the pronounced warmth at southern mid-to-high latitudes (Lunt et al.,



2013; Otto-Bliesner et al., 2017, 2021; Kageyama et al., 2021). This model-observation mismatch may be due to the models not capturing a prolonged period of meltwater influx from the Northern Hemisphere ice sheet melt, and subsequent weakening of the Atlantic Meridional Overturning circulation with warming of the south due to the bipolar seesaw effect (Holloway et al.,

2018; Sime et al., 2019a; Chadwick et al., 2023; Gao et al., 2024a).

From approximately 132 to 128 ka, in the period before peak LIG warmth, meltwater entered the North Atlantic from the deglaciation of the large Northern Hemisphere glacial-period ice sheets. The event is sometimes called Heinrich 11 (H11), although technically a Heinrich event is defined as the occurrence of very significant meltwater markers (*e.g.* ice-rafted debris) in North Atlantic sediment cores (Heinrich, 1988; Hemming, 2004), rather than being a simple label for the period marked by

Northern Hemisphere meltwater entering the oceans. The extra freshwater reduced the thermohaline circulation-driven part of the Atlantic Meridional Overturning Circulation (AMOC), and decreased the associated export and loss of global heat through and to the North Atlantic (Rahmstorf, 2000; Weaver et al., 2003; Clark et al., 2002; McManus et al., 2004). This resulted in large-scale changes in ocean circulation, including cooling in the North Atlantic and Arctic, and warming in the Southern and Global Ocean. Recent dating of the H11 event suggest that the meltwater could have persisted for 3000 to 5000 years

(length partly depending on definition of 'Heinrich') and that it is possible that the meltwater ceased directly before peak LIG temperatures at both poles (Marino et al., 2015; Stoll et al., 2022).

Given this progress on understanding the processes which are key to LIG climate at the poles, it seems timely to return to the question of the inference, or quantification, of polar warming during the LIG from ice core measurements. Ice core stable water isotopes ($\delta^{18}$O) from Antarctica and Greenland provide invaluable information on past variations of site temperature

(EPICA community members, 2004; Jouzel et al., 2007; Sime et al., 2013; Malmierca-Vallet et al., 2018; Domingo et al., 2020). However the retrieval of accurate LIG peak temperature information from ice core $\delta^{18}$O values is challenging (*e.g.* Jouzel et al., 2003; NEEM community members, 2013; Goursaud et al., 2021). In seminal work by Petit et al. (1999), isotopic measurements from the Vostok ice core in Antarctica demonstrated that the $\delta^{18}$O values closely correlate with temperature variations over the past 420,000 years, with a sensitivity of approximately 0.7‰/K; in West Antarctica, WAIS Divide Project Members

(2015) identified a regional sensitivity of 0.8‰/K; whilst for the NEEM core site in Greenland NEEM community members (2013) used 0.48 ± 0.1 ‰/K. The most important control on $\delta^{18}$O in precipitation in polar regions is site condensation temperature, thus the use of $\delta^{18}$O in precipitation to reconstruct condensation or site temperatures has been supported by these observed and simulated quasi-linear spatial relationships between isotopic ratios of surface snow and local surface temperature. However additional influences on $\delta^{18}$O can include changes in: the local boundary layer conditions (Krinner et al., 1997;

Noone and Simmonds, 2002); the seasonality of the precipitation (Sime et al., 2008, 2009); air mass trajectories and vapour to precipitation distance (Delaygue et al., 2000; Schlosser et al., 2004); evaporation and ocean surface conditions (Vimeux et al., 1999), particularly sea ice (Bertler et al., 2018; Holloway et al., 2016, 2017). These confounding issues make it imperative to study the relationships between $\delta^{18}$O and temperature for each climate shift, where we wish to understand temperature from measured $\delta^{18}$O changes in ice cores.

Isotope-enabled General Circulation Models (GCMs) allow a more direct comparison of model isotopic outputs with water isotope concentrations measured in natural archives, including ice cores. These models provide the most comprehensive means



that we possess to include all the factors that affect the $\delta^{18}O$ and surface air temperature changes (Werner et al., 2001; Werner and Heimann, 2002). They have therefore been used to investigate $\delta^{18}O$ and temperature in polar regions, and to provide a theoretical framework that accounts for both spatial and temporal variations in isotopic signals (Werner and Heimann, 2002; Sime et al., 2008). This study investigates water isotope changes in precipitation in the Antarctic and the Arctic regions that occurred between the PI and LIG using three isotope-enabled GCMs, and is the first isotopic-enabled multi-model study of the LIG. LIG simulations are run with each of these models according to the recent PMIP4 LIG protocol (Otto-Bliesner et al., 2017). These simulations are supplemented by a long 3000-year H11 experiment which is run using HadCM3. The Methods section (2) details these models, experimental set-ups, and ice core $\delta^{18}O$ and climate data, alongside analysis methods. Section 3 presents the results, Section 4 discusses the findings, and Section 5 concludes the study.

## 2   Methods

The Methods section details: the three water-isotope enabled models used; simulations run with these models; how model output is handled and analysed; and Antarctic and Greenland ice core $\delta^{18}O$ and wider set of observations used to evaluate the model simulations.

### 2.1   Models

The use of three isotope-enabled models aids in understanding model-dependency of results. The models are: the Max Planck Institute for Meteorology Earth System Model here referred to here as MPI-ESM-wiso (Cauquoin et al., 2019); the Goddard Institute for Space Studies coupled ocean-atmosphere model: GISS-E2.1-G (Kelley et al., 2020); and version 4.5 of the Hadley Centre Climate model: HadCM3 (Tindall et al., 2009). MPI-ESM (version 1.2 Giorgetta et al., 2013; Mauritsen et al., 2019) has been used for a wide range of paleoclimate simulations (*e.g.* Otto-Bliesner et al., 2021). Here we use the low resolution (LR) version, which has 1.875°×1.875° and 47 hybrid vertical levels in the atmosphere, and 1.5°×1.5° and 40 levels in the ocean component. Subsequent to the addition of water isotope module, is usually called MPI-ESM-wiso (Cauquoin et al., 2019). GISS-E2.1 is the Goddard Institute for Space Studies GISS ModelE2.1-G which has an atmospheric resolution of 2° latitude by 2.5° longitude, with 40 vertical layers up to 0.1 hPa coupled to the GISS Ocean v1 model with a resolution of 1° latitude by 1.25° longitude with 40 layers. Its vegetation uses the demographic global vegetation model called Ent Terrestrial Biosphere Model (Kelley et al., 2020). HadCM3 couples a hydrostatic atmospheric component HadAM3 and a barotropic ocean component HadOM3 (Tindall et al., 2009). The HadAM3 atmospheric model has a horizontal resolution of 2.5°×3.75° and 19 hybrid vertical levels, while the HadOM3 ocean model has a horizontal resolution of 1.25°×1.25° and 20 oceanic levels. The water isotope module of HadCM3 is described in Tindall et al. (2009).

### 2.2   Model simulations and output

Our series of simulations focuses on the LIG climate and isotope maximum in Antarctica and the Arctic. For each model, a pre-industrial (PI) control simulation is set-up following, or closely approximating the protocol in Eyring et al. (2016). For



**Table 1.** Simulation names, models versions, and boundary conditions. Note all simulations are run with a pre-industrial ice sheet configuration. The magnitude of the H11 freshwater forcing is 0.25 Sverdrups. See also Sections 2.1 and 2.2.

| Simulation name | Model (and version) | Orbital forcing | GHG forcing | Spin-up (years) | Duration of H11 forcing |
|---|---|---|---|---|---|
| ECHAM6$_{PI}$ | ECHAM6 (MPI-ESM-wiso) | | (CO$_2$=284 ppm, | 3500 | - |
| GISS$_{PI}$ | GISS (ModelE2.1-G) | 1850 C.E | CH$_4$=808 ppb, | 1000 | - |
| HadCM3$_{PI}$ | HadCM3 (UM4.5) | | N$_2$O=273 ppb) | 2000 | - |
| ECHAM6$_{127k}$ | ECHAM6 (MPI-ESM-wiso) | | (CO$_2$=275 ppm, | 2800 | - |
| GISS$_{127k}$ | GISS (ModelE2.1-G) | 127ka | CH$_4$=685 ppb, | 1000 | - |
| HadCM3$_{127k}$ | HadCM3 (UM4.5) | | N$_2$O=255 ppb) | 2000 | - |
| H11-200 | HadCM3 (UM4.5) | | (CO$_2$=275 ppm, | 1000 | 0.25Sv for 200 years |
| H11-3000 | HadCM3 (UM4.5) | 128ka | CH$_4$=705 ppb, | 3800 | 0.25Sv for 3000 years |
| H11-overshoot | HadCM3 (UM4.5) | | N$_2$O=265 ppb) | 3900 | 0.0Sv, 200 years post-H11-3000 |

each model, the PI experiment is run for a period long enough to reach a pseudo-equilibrium state (*i.e.*, negligible drifts in the atmosphere, surface, and mid depth ocean). For MPI-ESM-wiso, the simulation has been continued for 2500 model years

from a standard simulation without isotopes, which was run for 1000 years (so 3500 years in total) (Cauquoin et al., 2019). For GISS the pre-industrial simulation is begun from WOA observed conditions with isotopic composition of the whole ocean prescribed, and then spun-up for 1000 years, with the last 100 years used in this study. For HadCM3, the PI spin-up is more than 2000 years in length. Previous evaluation of each model against observations suggests that the simulated distribution of isotopes in polar precipitation compare well to present-day (*e.g.* Sime et al., 2008; Cauquoin et al., 2019; Oger et al., 2023).

As well as the PI control, for each model one LIG simulation is run, based on the Otto-Bliesner et al. (2017) Tier 1 lig127k protocol. The protocol specifies orbital and greenhouse gas concentrations fixed at 127-ka values (see Table 1). Atmospheric greenhouse gas concentrations are derived from ice core records at 127 ka (Otto-Bliesner et al., 2017), and use a CO$_2$ value of 275ppm. As specified in the protocol, sea level and ice sheet configurations are kept same as PI set-up for each model (Table 1). For MPI-ESM-wiso, the simulation has been continued for 1500 model years from a standard simulation without

isotopes, which was run for 1300 years (so 2800 years in total), starting from PI state for the isotopic spatial distribution in the ocean. For GISS, the LIG simulation has the same spin-up as for the PI. And for HadCM3, it was spun-up for 1000 years, starting from a spun-up PI. These long spin-ups mean that all simulations have reached pseudo-equilibrium states, as achieved for the PI simulation. As in Table 1, these simulations are referred to using the convention MODEL$_{TIMESLICE}$, for example, MPI-ESM-wiso$_{PI}$ and GISS$_{127k}$.

In addition to the primary multi-model 127k simulations, we use HadCM3 to run a Heinrich Stadial 11, or H11, simulation which directly extends the Holloway et al. (2018) simulation. This enables exploration of the effects of the Heinrich 11 (or Termination 2) ice sheet melt event, which preceded 127 ka, and how the associated meltwater entering the North Atlantic



on climate, and associated $\delta^{18}O$ changes. The H11 simulation is run on from a 128 ka simulation, spun-up for the orbital and GHG conditions of 128ka over 800 years. Our Holloway et al. (2018) H11 experiment has orbital and GHG forcing that are for 128ka rather than 127ka. Gao et al. (2024a) shows that standard 127-ka and 128-ka HadCM3 simulations give very similar results: orbital changes between 127 and 128ka are slight; whereas the impacts of the meltwater forcing are large. Our H11 simulation continues to apply the same 128 ka orbital and GHG conditions, with the addition of meltwater, via standard hosing-type set-up, to the North Atlantic. In support of this variant, Otto-Bliesner et al. (2017) introduced a Tier 2 sensitivity experiment, which similarly includes a prescribed meltwater flux to the North Atlantic. Since Holloway et al. (2018); Guarino et al. (2023); Gao et al. (2024a) suggest that 3000 years of this meltwater forcing is required to allow the impacts of northern hemisphere ice sheet melt on LIG climate to fully manifest in the Southern Ocean and Antarctica, 0.25Sv of melt water is added uniformly to the surface North Atlantic Ocean between 45°and 70°N for 3100 years.

The meltwater forcing is then stopped, and an additional simulation is branched off at 3000 years, where orbital and GHG forcing continued to be applied but no further meltwater is added. This extra branch is useful to capture the initial effects of post H11 recovery, sometimes referred to as the AMOC overshoot. As in Table 1, these simulations are referred to as H11-TIME/BRANCH.

The H11 simulation represents a significantly longer meltwater forcing than previous LIG simulations run with the HadCM3 model (Stone et al., 2016; Holloway et al., 2018). Given the actual meltwater isotopic composition from deglaciating Northern Hemisphere Ice Sheets is highly uncertain, following Holloway et al. (2018) the ocean isotopic composition for the H11 simulation is set to 0 ‰ (equivalent to present day) to ensure no negative drift in whole ocean isotopic composition. This effectively uses the assumption that by the end of Heinrich Stadial-11 sea level has reached approximately present-day values, resulting in a globally integrated ocean isotopic composition of 0 ‰.

### 2.2.1 Model output: calculation of anomalies

All climatologies for LIG simulations are calculated using the final 100 years of output and are compared to the PI climatologies from each individual model. For the H11 simulation, specific periods are picked out for analysis at 200 and 3000 years into H11 forcing, and post H11, or AMOC overshoot; referred to respectively as H11-200, H11-3000, and H11-overshoot (See Figure A2, Table 1). For the H11 simulation, climatology for 100 year slices from initial Phase (from years 200-300 after hosing started), final phase (year 3000-3100) and part of the overshoot phase (year 200-300, after hosing is stopped) are computed. Each simulation is summarised in Table 1. LIG - PI anomalies are generally presented using the $\Delta$ notation. So, for example, anomalies in Surface Air Temperatures (SATs) would be denoted $\Delta$SAT and be calculated for MPI-ESM-wiso as MPI-ESM-wiso$_{127k}$-MPI-ESM-wiso$_{PI}$, or for year 3000 of the H11 simulation using [H11-3000] -[HadCM3$_{PI}$].

Bilinear interpolation in latitude–longitude space is used to extract values at the ice core locations from the gridded model output. The time mean variables [and units], or climate quantities, used for each model are: surface air temperature (SAT in K), precipitation (Precip in mm/month), sea surface temperature (SST, K), sea ice concentration (SIC in %), and sea ice area (SIA in mill. km$^2$). To extract model output at each site, nearest neighbour interpolation is used. Summer SST is defined as the average January-February-March SST. SIA is estimated by summing the product of Antarctic SIC and grid cell area.



All $\delta^{18}O$ values are mass-weighted and presented in parts per thousand (‰), and calculated:

$$\delta^{18}O = \left( \frac{\left(\frac{^{18}O}{^{16}O}\right)_{sample}}{\left(\frac{^{18}O}{^{16}O}\right)_{standard}} - 1 \right) \times 1000 \tag{1}$$

### 2.2.2 Model output: $\delta^{18}O$ drivers quantifying seasonality of $\delta^{18}O$ and precipitation

To help quantify the impacts of $\delta^{18}O$ and precipitation seasonality on the $\delta^{18}O$ changes between PI and LIG, we show a simple breakdown of the drivers of $\Delta\delta^{18}O$ following the method used in Holloway et al. (2016), Sime et al. (2019b), and Oger et al. (2023):

$$\Delta\delta^{18}O_{seasonal} = \frac{\sum_j \delta^{18}O_j^{LIG} \times Precip_j^{PI}}{\sum_j Precip_j^{PI}} - \frac{\sum_J \delta^{18}O_j^{PI} \times Precip_j^{PI}}{\sum_j Precip_j^{PI}} \tag{2}$$

$$\Delta Precip_{seasonal} = \frac{\sum_j \delta^{18}O_j^{PI} \times Precip_j^{LIG}}{\sum_j Precip_j^{LIG}} - \frac{\sum_j \delta^{18}O_j^{PI} \times Precip_j^{PI}}{\sum_j Precip_j^{PI}} \tag{3}$$

The summations (j) are over the 12 months of a year, using monthly climatological values.

### 2.2.3 Model output: uncertainties and $\delta^{18}O$ versus SAT relationships

We use linear regression to investigate the relationship between simulated $\Delta\delta^{18}O$ and $\Delta SAT$. The regression is fitted using one and two part fits. In the first case, this implies no change in SAT if there is no change in $\delta^{18}O$, and *vice versa*, whilst in the second case changes in SAT and $\delta^{18}O$ can be partly independent. This second case would take account of cases where, for example, there a PI-to-LIG climate change that affects only SAT or only $\delta^{18}O$, but not both. All linear regressions are performed using the ordinary least squares (OLS) method, utilizing the Python package 'statsmodels' (Seabold and Perktold, 2010), which provides robust tools for linear modeling. To quantify the uncertainties in the predicted dependent variable, confidence intervals are calculated at the 95% level. These uncertainties help in understanding the reliability of fit. This is important given there are only six data points (corresponding to six model simulations) available. This limited number of simulations (data points) inherently increases the uncertainty of the fit. Uncertainties are calculated using the standard error of the regression coefficients, which takes into account the variance of the residuals and the distribution of the independent variable.

### 2.3 Arctic and Antarctic ice core $\delta^{18}O$ data

Ice core data for the LIG is available from both Greenland (Arctic) and Antarctica. LIG ice layers have been found in deep Greenland ice cores. Although dating these layers presents challenges and some may be missing, Domingo et al. (2020) inferred the "most likely" LIG peak $\delta^{18}O$ values along with the associated uncertainty ranges from six deep ice cores (Figure A1). These



are: NEEM (NEEM community members, 2013); NGRIP (Members, North Greenland Ice Core Project, 2004); GRIP (Landais et al., 2003); GISP2 (Suwa and Bender, 2008; Yau et al., 2016); Camp Century (Johnsen et al., 2001); and DYE3 (Johnsen et al., 2001). These data are re-used in this study (Table A1, first four columns upper half).

Goursaud et al. (2021) presented data from six published East Antarctic ice core records that provide $\delta^{18}$O measurements from the PI and LIG (Table A1, first four columns lower half). Most of the $\delta^{18}$O values were in practice extracted from Bazin et al. (2013) or Masson-Delmotte et al. (2011). The sites and key original references for each site (Figure A1) are: Vostok (Petit et al., 1999); Dome F (Kawamura et al., 2007); EDC (Jouzel et al., 2007); EDML (EPICA Community Members, 2006); TALDICE (Stenni et al., 2011); and Taylor (Steig et al., 2000). The observed PI value corresponds to the average over the period 1850-1900, and the LIG value corresponds to the maximum value during the LIG period (Goursaud et al., 2021). Of these cores, Taylor is least confidently dated. It receives less emphasis in the results.

## 2.4 Overview of Global, Arctic, and Southern Ocean LIG temperature and sea ice changes

Given sea surface conditions and global climate play a key role in setting SAT and $\delta^{18}$O in polar regions, it is also helpful to evaluate the model simulations against global-scale observations from the LIG. An overview of these observation-sets are provided here. Hoffman et al. (2017) estimate a global annual sea surface temperature warming during the LIG of 0.5±0.3K. Based largely on this estimate, Fischer et al. (2018) then estimated a global mean maximum surface air temperature LIG warming of 0.8±0.5K above pre-industrial levels. This also seems to be in line with more recent estimates of LIG global mean ocean temperature warming of 1.1±0.3K at 129ka, decreasing to ~0K by around 124ka (Shackleton et al., 2020). In the Arctic, Sime et al. (2023); Guarino et al. (2020) and Vermassen et al. (2023) show together that the Arctic was likely occasionally or often practically sea ice free at 127ka, *i.e.* that the total SIA in the Arctic was less than 1 mill. km$^2$, and that the summer ΔSAT in the Arctic was +3.7±1.5K warmer relative to the PI, north of 70N. For the Southern Ocean, sea ice at 127 ka was reduced by 40-60% over winter compared to the preindustrial (Holloway et al., 2016, 2017; Chadwick et al., 2023), and Gao et al. (2024a) alongside Capron et al. (2017); Hoffman et al. (2017); Chandler and Langebroek (2021); Chadwick et al. (2021) estimate a Southern Ocean annual SST anomaly of around +1.3°C at 127 ka relative to present.

## 3 Results

We first assess the global and polar sea surface changes, with a focus on summer ΔSAT and ΔSIC for the Arctic and Antarctic regions to assess whether simulations capture the observed 127ka warming. The next portion of the results considers ΔSAT, ΔPrecip, and Δ$\delta^{18}$O across Greenland and Antarctica, alongside an evaluation of the impact of changes in seasonality on Δ$\delta^{18}$O. Then we use simulated Δ$\delta^{18}$O and ΔSAT to infer ΔSAT from Δ$\delta^{18}$O measured in ice cores, and to obtain estimates of the climate-related polar warming at Greenland and Antarctic ice core sites.





### 3.1 Model climatology: Do our simulations capture the polar and global climate of 127ka?

#### 3.1.1 Global temperature

The 127k simulations show a small global cooling with global-mean $\Delta$SAT from -0.23 to -0.11 K and $\Delta$SST from -0.17 to -0.07K (Table A2). In association with this, HadCM3$_{127k}$ and MPI-ESM-wiso$_{127k}$ show a cooling of -0.016 to -0.045K in
global mean ocean temperature (GMOT). The longer H11 simulations show progressive global warming during the H11 event: the H11-200, H11-3000, and H11-overshoot simulations show global-mean $\Delta$SAT of -0.23, 0.35, and 1.09 and $\Delta$SST of 0.04, 0.45, and 0.83 K, and $\Delta$GMOT for the H11-3000 and H11-overshoot simulations are both +0.8K (Table A2). Thus whilst all 127k simulations show a slight global cooling, the H11 simulations (particularly the H11-3000) show warmer global SST, SAT, and GMOT changes. Towards the end of the 3000 years of H11 forcing, results from these H11 simulations match the Fischer
et al. (2018) and Shackleton et al. (2020) estimates of these global quantities.

#### 3.1.2 Arctic temperature and sea ice

Kageyama et al. (2021) and Sime et al. (2023) show that all PMIP4 127k simulations have a decrease in summer SIA between the PI and 127k. However the simulations of present day and pre-industrial Arctic SIA is very variable between models, and often does not agree with observations. This can strongly affect subsequent assessment of Arctic $\Delta$SAT, $\Delta$SST, and $\Delta$SIA. We
find similar results for our three 127k simulations (Figure 1).

The three 127k simulations show in the Arctic a summer $\Delta$SAT of +1.7 to +2.7 K and a summer $\Delta$SIA reduction of between 35-55% (Table A3). Of the three models, MPI-ESM-wiso and HadCM3 have an absolute 127k summer SIA of 1.7-2.7 mill. km$^2$, which is not too far from the expected LIG 127ka sea ice state (Sime et al., 2023), however, the summer $\Delta$SAT of +1.7 to +1.8K is roughly half of the observed +3.7$\pm$1.5K warming (Sime et al., 2023; Guarino et al., 2020). Thus these simulations
underestimate both actual 127k summer sea ice loss, and associated Arctic summer warming. The 127k simulation that warms the most, the GISS model, has a simulation of both PI and 127k Arctic SIA that is much too large, which makes interpretation of the GISS$_{127k}$ Arctic changes difficult (*c.f.* Kageyama et al., 2021): effectively whilst there is a huge loss of sea ice from the LIG to the PI which drives a larger +2.7k warming of $\Delta$SAT; 6.4 mill. km$^2$ remains at 127ky, which is more than five times the area of the most likely 127ky SIA (Sime et al., 2023). This vast extra area of simulated SIA strongly affects SAT, precipitation
and $\delta^{18}$O, making it more difficult to draw conclusions from this particular GISS simulation. It is easier to draw conclusions from the MPI-ESM-wiso$_{127k}$ and HadCM3$_{127k}$ simulation results for the Arctic, whilst bearing in mind that these still show only half the expected summertime $\Delta$SAT. In the case of HadCM3$_{127k}$ the too-small warming is likely partly due to the PI featuring too little ice, so that summer $\Delta$SIA is relatively small at a 41% reduction (Table A3). For MPI-ESM-wiso$_{127k}$, the PI SIA and $\Delta$SIA is likely a little better, but nevertheless it also shows only half of the expected summer $\Delta$SAT (Table A3).
Of the H11 HadCM3 simulations (Figure 2), both H11-200 and H11-3000 show substantial annual mean growth of sea ice and colder sea surface and air temperatures across the Arctic and North Atlantic: Arctic annual-mean $\Delta$SAT is -0.3K, -2.7K and -1.5K at years 0 (HadCM3$_{127k}$), 200 (H11-200), and 3000 (H11-3000) of the H11 meltwater forcing, although due to the orbital forcing Arctic summer $\Delta$SAT is +1.7, +0.8, and +1.2K at these years (Table A3). However, during the AMOC overshoot, the

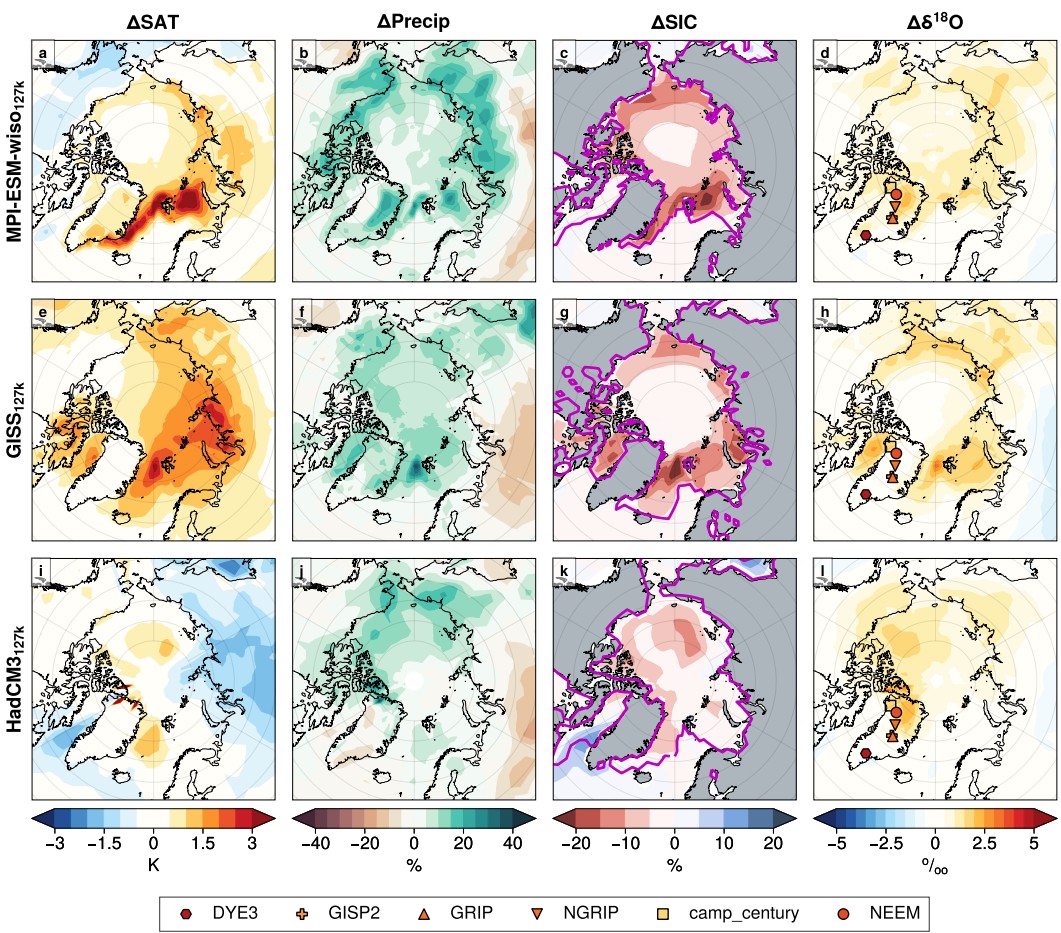

**Figure 1.** Anomalies in surface air temperature (ΔSAT), precipitation (ΔPrecip), sea ice concentration(ΔSIA) and $\delta^{18}O$ over Arctic region for the 127k simulations from the PI. See Table 1 for simulation details. The shaded symbols in right column shows ice core $\Delta\delta^{18}O$ values as described in Section 2.3 and Table A1. Precipitation differences are expressed as percentage deviation from mean PI values. The contours in Column 3 indicates the sea ice extent in each PI simulation.



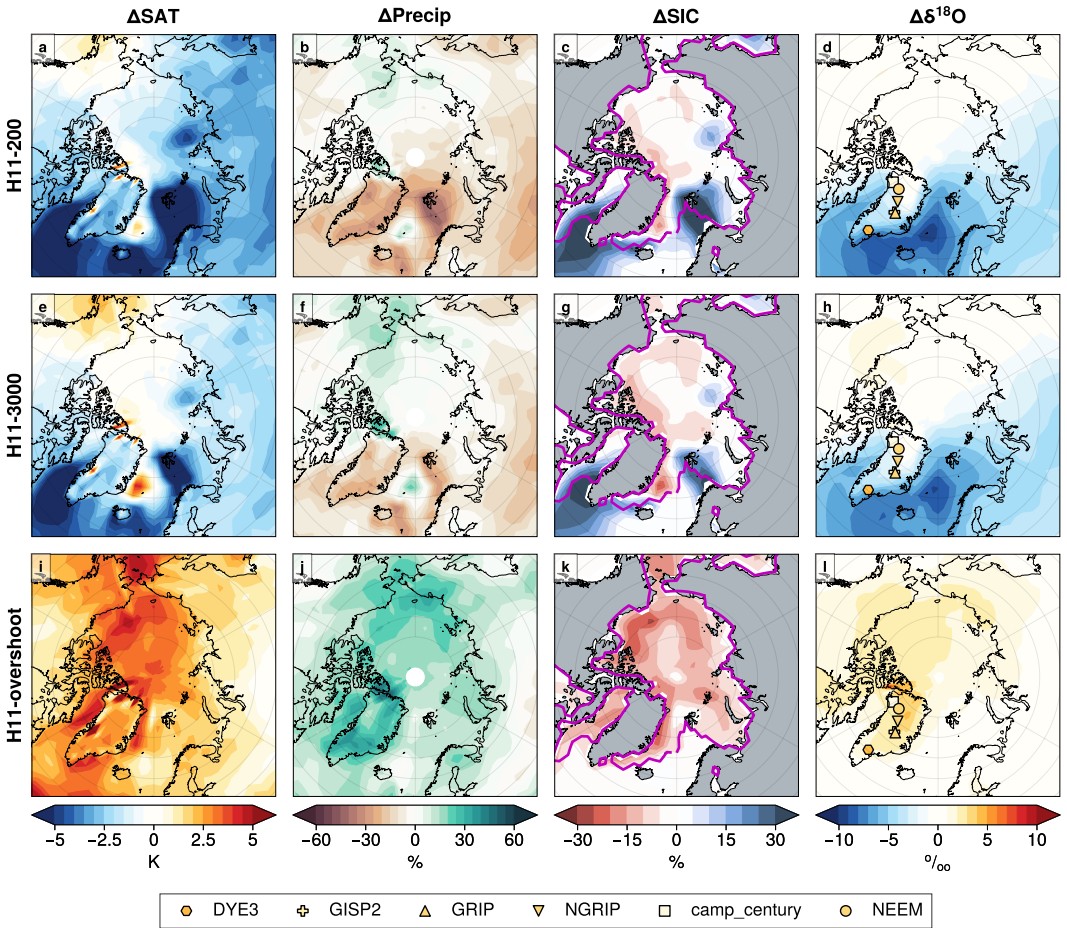

**Figure 2.** $\Delta$SAT, $\Delta$Precip, $\Delta$SIC and $\Delta\delta^{18}$O over the Arctic region for the H11 simulations compared to PI; all columns and rows as for Figure 1. Note range of colorbars is wider than in Figure 1.

Arctic becomes very much warmer. H11-overshoot shows an Arctic annual-mean $\Delta$SAT of +2.2K, summer $\Delta$SAT of +3.0K, and a practically sea-ice free Arctic summer SIA of 0.7 mill. km$^2$, due to the rapid resumption of the AMOC and advection of warm ocean waters into the Arctic.

Thus, of our six simulations, only H11-overshoot captures the majority of the observed Arctic warming: 81% of the summer Arctic $\Delta$SAT; a 75% loss of summer SIA; and an occasionally sea-ice free Arctic.

### 3.1.3 Southern Ocean and Antarctic region temperature and sea ice

Comparison against the recent synthesis in Gao et al. (2024a) allows us to ascertain the percentage of Southern Ocean $\Delta$SST and $\Delta$SIA captured by the simulations (Figure 3 and 4; Table A4). The three 127k simulations capture no more than 12 % of the 127ka Southern Ocean warming (Figure 5). Even MPI-ESM-wiso$_{127k}$, which shows a relatively large annual Antarctic





**Figure 3.** Reconstructed climate anomalies : (a) annual ΔSST and ΔSAT; (b) summer ΔSST; and (c) September ΔSIC over the Antarctic and Southern Ocean region for the 127k simulations compared to each PI simulation. Gao et al. (2024a) compiled these datasets from Capron et al. (2017); Hoffman et al. (2017); Chandler and Langebroek (2021); Chadwick et al. (2021).

reduction of ΔSIA of 23% and a September sea ice (winter maximum) that is reduced by 63 % (MPI-ESM-wiso generally shows the largest 127k simulation drop in all SIA quantities in both hemispheres), shows little Southern Ocean warming in 
association with its sea ice loss (Figure 3ab; Table A4).





**Figure 4.** As Figure 3 but for the H11 simulations compared to the PI simulation.

In contrast, the H11-3000 simulation captures most of the warming and sea ice loss in Gao et al. (2024a) (Figure 4; Table A4). South of 40°S, annual-mean ΔSST rises by 1.3K, while reconstructed average anomalies range from +2.2K to +2.7K; summer ΔSST is 1.1K, close to 1.2-2.2K reconstructed average anomalies; September (winter) SIA reduces by 40%, similar to reconstructed 40% reduction of Southern Ocean SIC (Gao et al., 2024a). This agreement between the H11-3000 Southern and Global Ocean warming (Section 3.1.1) tends to support the idea that it is necessary to take account of the affects of the long





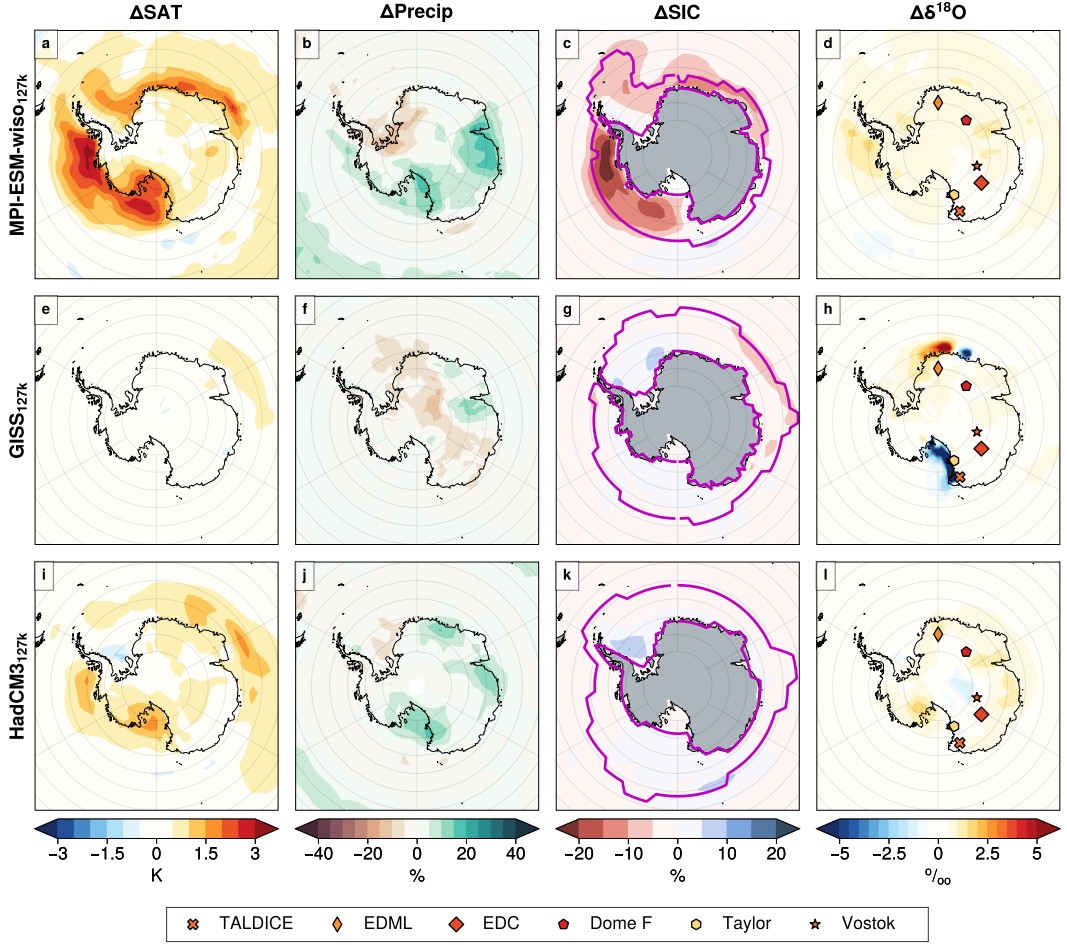

**Figure 5.** Anomalies in SAT, Precip, SIC and $\delta^{18}O$ over the Antarctic and Southern Ocean region for the 127k simulations compared to PI; all columns and rows as for Figure 1.

H11 meltwater event to allow simulations to capture 127k changes in the Southern Ocean and Antarctic region (Otto-Bliesner et al., 2017; Holloway et al., 2018; Sime et al., 2019a).

Overall, the 127k simulations do not capture the observed Southern Ocean warming and sea ice loss, but the H11-3000 simulation captures between 50-100% of the warming (Table A4), depending on the particular synthesis (Gao et al., 2024a) 280 and observation type.





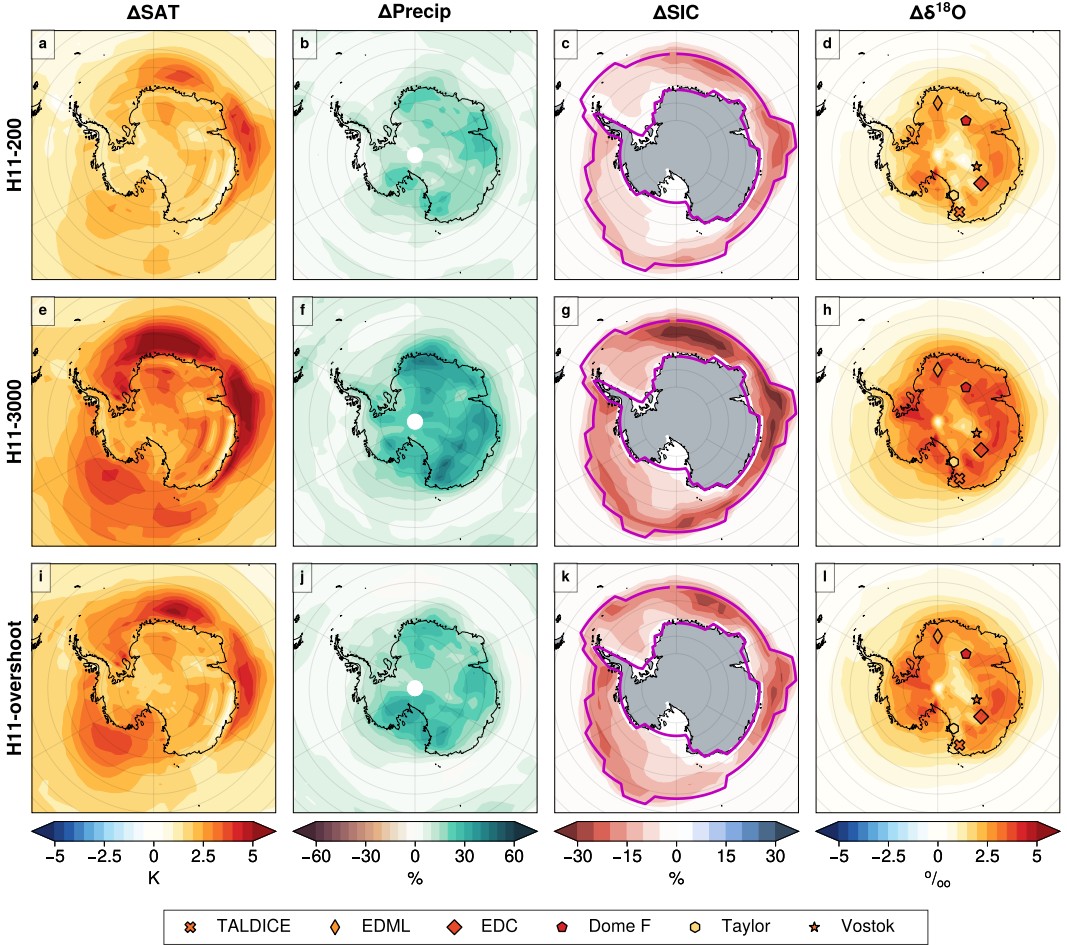

**Figure 6.** Anomalies in SAT, Precip, SIC and $\delta^{18}$O over the Antarctic and Southern Ocean region for the H11 simulations; all columns and rows as for Figure 1. Note range of colorbars is wider than in Figure 5, except for $\delta^{18}$O.

## 3.2 Changes over the ice-sheets: $\Delta$SAT, $\Delta$Precip, and $\Delta\delta^{18}$O

Having looked at how the simulations capture, or not, Global, Arctic and Southern Ocean $\Delta$SST, $\Delta$GMOT, $\Delta$SAT, and $\Delta$SIA, we now move on to look more closely at the simulation of $\Delta$Precip, and its associated $\Delta\delta^{18}$O values over the Greenland and Antarctic ice sheets, with a focus on ice core locations.

### 285  3.2.1  Greenland

Most of the 127k simulation $\Delta$SAT, $\Delta$Precip, and $\Delta\delta^{18}$O changes are strongly associated with sea ice changes, including $\Delta$SIA but also the initial (PI) sea ice state, particularly the summertime changes (Section 3.1.2; and Figure 1).





GISS$_{127k}$ has more than 6 mill. km$^2$ of Arctic SIA left in the summer, meaning there is little open water or associated opportunity for proximal sea evaporation in summer. It therefore shows very little change in $\Delta$SAT, $\Delta$Precip, and $\Delta\delta^{18}$O over
Greenland (Figure 1): and its Greenland core-mean $\Delta\delta^{18}$O value is 0.0‰. In contrast, because HadCM3$_{127k}$ and MPI-ESM-wiso$_{127k}$ have summertime 127k SIA's of 1.7 and 2.7 mill. km$^2$ their $\Delta\delta^{18}$O are enriched by +0.6 ‰ and +1 ‰, respectively. This is still considerably smaller than the observed core-mean $\Delta\delta^{18}$O of +3.2‰ (Table A1). Thus the mean Arctic summertime $\Delta$SAT is about half of the observed value, but the Greenland core-mean $\Delta\delta^{18}$O values are only one quarter to one third of the observational estimates (Domingo et al., 2020; Sime et al., 2023).

There are also variations across Greenland in $\Delta$Precip and $\Delta\delta^{18}$O between the 127 k simulations. For example, HadCM3$_{127k}$ tends to be drier across southern Greenland, but the isotopic response over northern Greenland is much stronger, even when the precipitation increase there is weaker compared to MPI-ESM-wiso and GISS. This could be indicative of more detailed source and transport related response in $\delta^{18}$O. Although outside the scope of this manuscript, further water tracer-simulations might be useful to diagnose these (*e.g.* Gao et al., 2024b).

For the H11-200 and H11-3000 simulations $\Delta\delta^{18}$O is depleted over Greenland and the wider northern Atlantic sector, commensurate with the mean-annual SIA increase and cooling (Section 3.1.2; and Figure 8). However, the warm Arctic and summer sea ice free Arctic of H11-overshoot leads to strongly enhanced precipitation and reduced sea ice, all of which contributes to the very substantial H11-overshoot increase over Greenland core-mean $\Delta\delta^{18}$O of 2.8 ‰, compared to a Greenland core-mean $\Delta\delta^{18}$O of 3.2 ‰: *i.e.* H11-overshoot captures 81-87% of both summer Arctic $\Delta$SAT and Greenland core-mean $\Delta\delta^{18}$O.

### 3.2.2 Antarctica

Section 3.1.3 indicated that the 127k simulations do not capture much of the observed Southern Ocean region warming. So in turn, the Antarctic $\Delta$SAT, $\Delta$Precip, and $\Delta\delta^{18}$O 127k simulation changes are also very small, and not consistently in the correct direction relative to Antarctic core site 127ka measurements (Figure 5; Table A1): core-mean $\Delta\delta^{18}$O results are +0.0, -1.0, and +0.3 ‰ for HadCM3$_{127k}$, GISS$_{127k}$ and MPI-ESM-wiso$_{127k}$, respectively. This is compared with a observed core-mean $\Delta\delta^{18}$O
value of +3.3 ‰. This is somewhat like the Greenland 127k results; which also indicate that the mismatches with observed ice core $\Delta\delta^{18}$O seem mostly driven by simulation of sea ice and SST changes, or lack thereof.

Within Antarctica, of the three 127k simulations, the MPI-ESM-wiso$_{127k}$ features the strongest Antarctic surface warming, including a strong warming around the west side of the Antarctic Peninsula and over other coastal regions. This likely is because it does show some 127k simulation Antarctic sea ice loss. The sea ice loss also leads to rather uniform $\delta^{18}$O en-
richment across the Antarctic and surrounding ocean. GISS$_{127k}$ and HadCM3$_{127k}$ show different patterns of $\Delta\delta^{18}$O across Antarctica. HadCM3 simulates slight enrichment of $\delta^{18}$O towards inland Antarctic, while GISS$_{127k}$ simulates strong enrichment around the Ross sea sector. Unlike MPI-ESM-wiso$_{127k}$, GISS$_{127k}$ and HadCM3$_{127k}$ do not show significant sea ice reduction, although HadCM3$_{127k}$ does simulate surface warming over Antarctic and coastal oceans, but not as strong as MPI-ESM-wiso$_{127k}$. GISS$_{127k}$ does not show any much change in $\Delta$SAT. Both MPI-ESM-wiso$_{127k}$ and HadCM3$_{127k}$ have are wetter
(larger $\Delta$Precip) over Ross sea sector. MPI-ESM-wiso$_{127k}$ is drier across Weddell sea to Dronning Maud Land, while GISS$_{127k}$





is drier inland, with only exception over East Antarctic from around 60-100 °E. It is difficult to know what, if anything, to draw from these multi-model variations.

Over Antarctica the H11 simulations show relatively uniform $\Delta$SAT, $\Delta$Precip, and $\Delta\delta^{18}$O (Figure 6). The H11 Antarctic core-mean values for: $\Delta$SAT are +0.5K, +2.0, +3.0, and +2.4K at 0, 200, 3000 years and for the overshoot; and similarly of core-mean $\Delta\delta^{18}$O values of +0.0, +1.9, 3.1, and +2.3 ‰ (Table A1; Figure 6), compared to the observed core-mean $\Delta\delta^{18}$O value of 3.3 ‰. This means that H11-3000 simulation captures 94% of the observed Antarctic core-mean $\Delta\delta^{18}$O. This is commensurate with the previous sections which also show that the longer H11-3000 simulation also captures 50-100% of the Southern Ocean warming and sea ice loss.

### 3.3  The impact of changes in seasonality on $\Delta\delta^{18}$O

Decomposition of the impact of changes in seasonality on $\Delta\delta^{18}$O can be useful to better understand drivers of $\Delta\delta^{18}$O. See Section 2.2.2 for a description of the breakdown of $\Delta\delta^{18}$O into the two $\Delta\mathrm{Precip}_{seasonal}$ (precipitation seasonality changes) and $\Delta\delta^{18}\mathrm{O}_{seasonal}$ ($\delta^{18}$O seasonality, or source effect, changes) that is used here.

#### 3.3.1  Arctic

For the 127k simulations, precipitation source related ($\Delta\delta^{18}\mathrm{O}_{seasonal}$) changes in the Arctic, particularly for MPI-ESM-wiso$_{127k}$ and GISS$_{127k}$ contribute strongly to the enrichment in $\delta^{18}$O in regions of large sea ice loss; where there is increased exposure of the ocean surface there will be more (local) evaporation (Figure 7, first and third column). Given the large summer losses of Arctic sea ice, we would also expect to see an increase in summer precipitation for some simulations. Likely due their small Arctic SIAs, HadCM3$_{127k}$ and MPI-ESM-wiso$_{127k}$, do show the largest contribution from seasonal shift of precipitation (increase of warm season precipitation, $\Delta\mathrm{Precip}_{seasonal}$), on $\Delta\delta^{18}$O over Arctic. Indeed for HadCM3$_{127k}$ most increase in $\Delta\delta^{18}$O is associated with these $\Delta\mathrm{Precip}_{seasonal}$ precipitation seasonality changes. For MPI-ESM-wiso, the contributions of $\Delta\mathrm{Precip}_{seasonal}$ and $\Delta\delta^{18}\mathrm{O}_{seasonal}$ are more similar. Figure 1 and A3-A7 depict how $\Delta\mathrm{Precip}_{seasonal}$ changes in these models are depend largely on the loss of summertime high Arctic sea ice, which drives an increase in (local) warm season precipitation (Figure 7, middle column).

There is also some relationship between the two separate seasonality terms: the increased $\Delta\delta^{18}\mathrm{O}_{seasonal}$ term does tend to contribute to the enrichment in $\delta^{18}$O where $\Delta$Precip also increases (Figure 1), indicating that rainout and distance to source effects on $\delta^{18}$O are sometimes closely related. As would be expected, where $\Delta$Precip decreases $\Delta\delta^{18}\mathrm{O}_{seasonal}$ also tends to decrease.

For the H11 simulations, although HadCM3$_{127k}$ and H11-overshoot both show a warmer Arctic, the decomposition of $\Delta\delta^{18}$O for each simulation shows distinctly different behaviour (Figure 10). As above, whilst for HadCM3$_{127k}$ most increase in $\Delta\delta^{18}$O is associated with these $\Delta\mathrm{Precip}_{seasonal}$, this is completely reversed for H11-overshoot, with almost all of the increase driven by $\Delta\delta^{18}\mathrm{O}_{seasonal}$, rather than $\Delta\mathrm{Precip}_{seasonal}$. This is because there is a large annual-mean, including wintertime, loss of sea ice in H11-overshoot, driven by the increased AMOC and global ocean temperature (rather than purely the orbital 127k spring-summer losses), leading to much more winter time, and closer sourced wintertime, precipitation over Greenland.

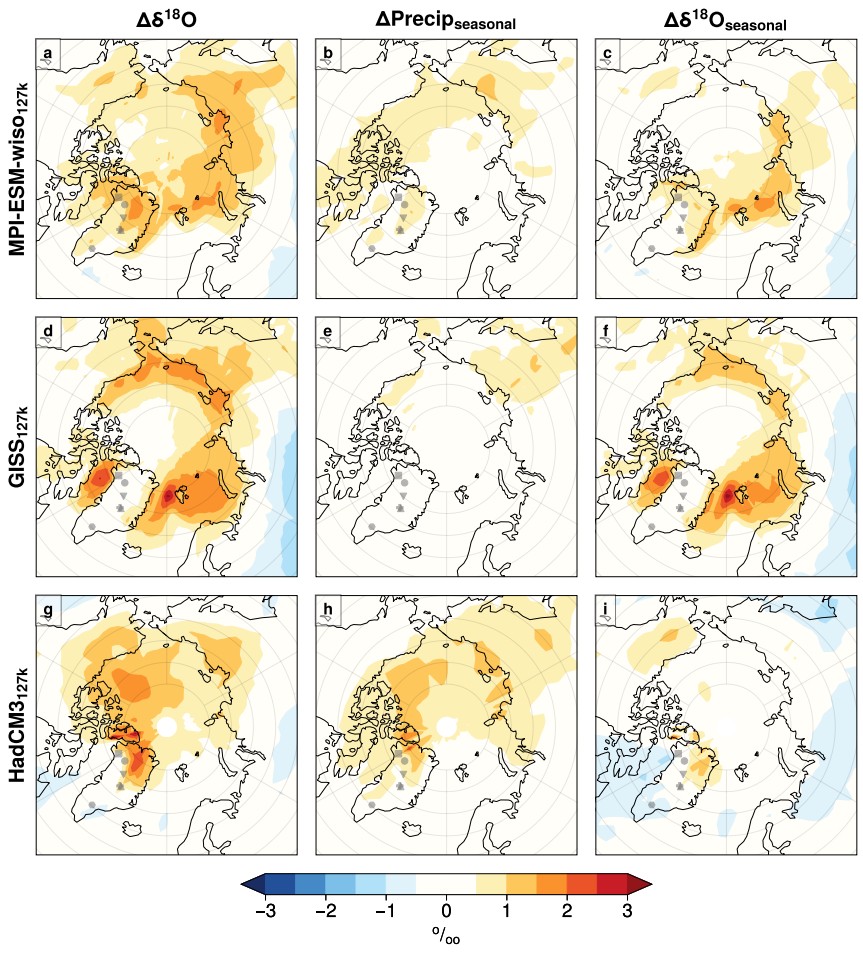

**Figure 7.** The impact of seasonality change on $\Delta\delta^{18}O$: decomposition of $\Delta\delta^{18}O$ values into contribution from seasonal variability into precipitation ($\Delta Precip_{seasonal}$) and source variability ($\Delta\delta^{18}O_{seasonal}$) in the Arctic for the 127k simulations. See Section 2.2.2 for full details.



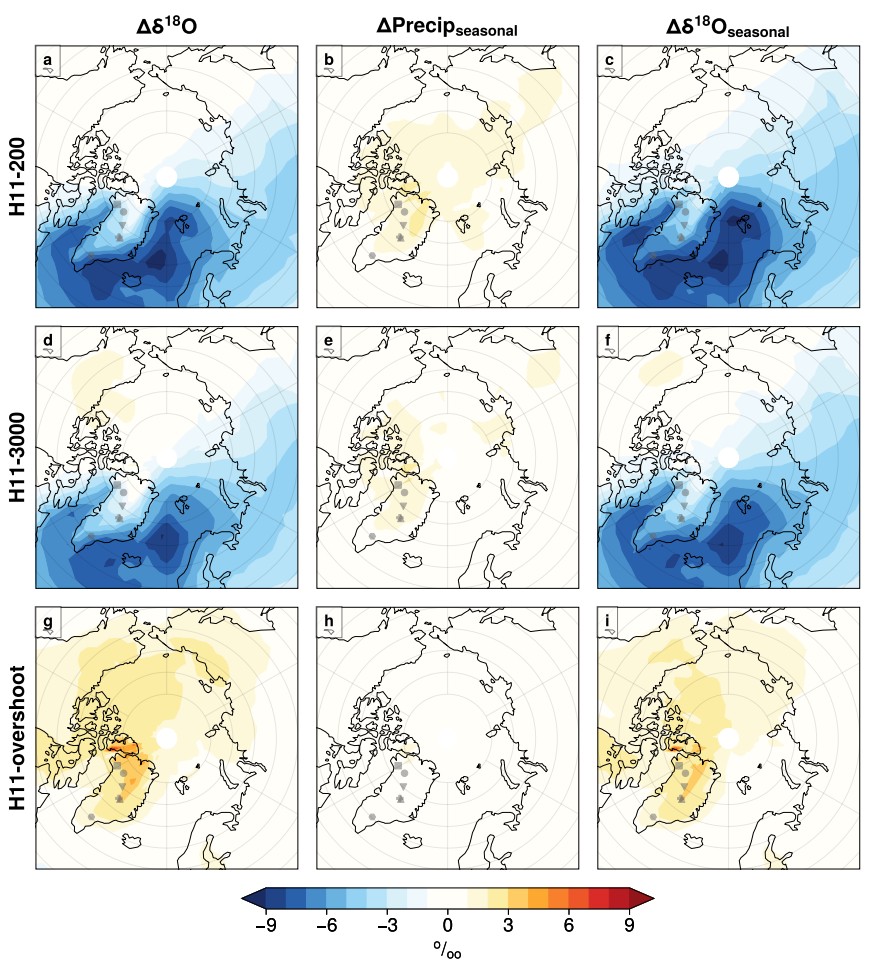

**Figure 8.** The impact of seasonality change on $\Delta\delta^{18}$O: : decomposition of $\Delta\delta^{18}$O values into contribution from seasonal variability into precipitation ($\Delta Precip_{seasonal}$) and source variability ($\Delta\delta^{18}O_{seasonal}$). Differences between the PI and H11 simulations, all columns and rows as for Figure 7.





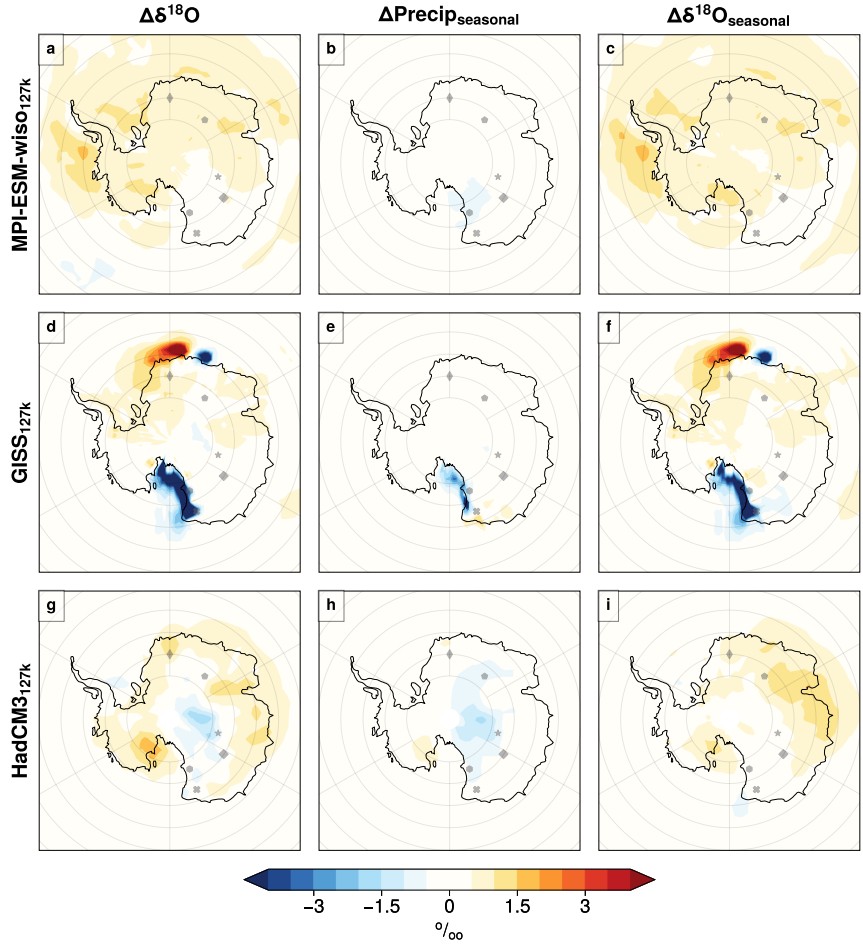

**Figure 9.** The impact of seasonality change on $\Delta\delta^{18}O$: decomposition of $\Delta\delta^{18}O$ values into contribution from seasonal variability into precipitation ($\Delta\mathrm{Precip}_{seasonal}$) and source variability ($\Delta\delta^{18}O_{seasonal}$). Differences between the PI and 127k simulations for the Antarctic region, all columns and rows as for Figure 7.

### 3.3.2 Antarctica

For the mainly orbitally-driven 127k simulations in Antarctica, precipitation seasonality changes ($\Delta\mathrm{Precip}_{seasonal}$), contribute to depletion in $\delta^{18}O$ over Antarctica (Figure 9, middle column). For simulations where overall precipitation is increasing/not changing, including HadCM3$_{127k}$ and for the Ross Sea/East Antarctic sector in MPI-ESM-wiso$_{127k}$, this can be explained by relative increase in precipitation that falls during colder seasons during the 127k simulations relative to the PI, leading to reduced $\delta^{18}O$. However, opposing this, sea-ice retreat and shorter source-to-site vapour transport pathways can 360 raise $\delta^{18}O$ values (Figure 9, right column). Together these two terms (the seasonality of precipitation, and seasonality of $\delta^{18}O$




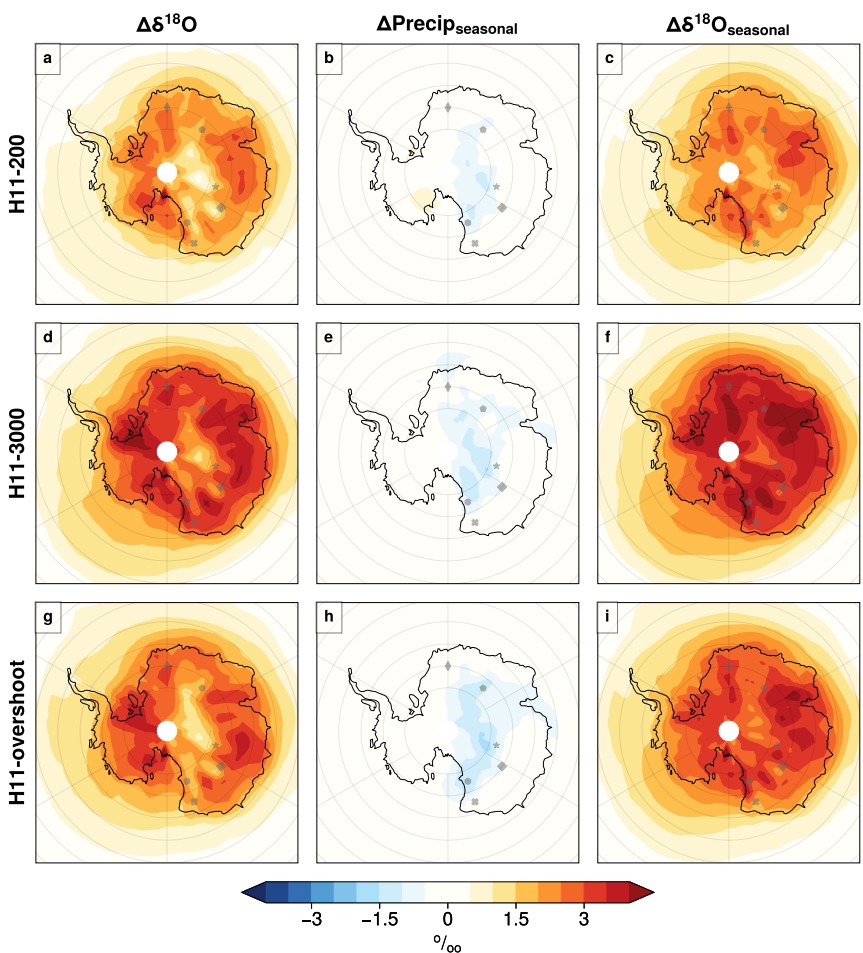

**Figure 10.** The impact of seasonality change on $\Delta\delta^{18}$O: decomposition of $\Delta\delta^{18}$O values into contribution from seasonal variability into precipitation ($\Delta Precip_{seasonal}$) and source variability ($\Delta\delta^{18}O_{seasonal}$). Differences between the PI and H11 simulations for the Antarctic region, all columns and rows as for Figure 7.



changes) can therefore lead to rather small, sometimes near net near-zero simulated $\Delta\delta^{18}$O across Antarctica between the LIG to PI for the 127k simulations (Figure 9).

The GISS$_{127k}$ simulation has a very unique pattern of $\delta^{18}$O change in Antarctic with extreme low values over the Ross Sea sector and a very local pattern of low and high values in East Antarctic coastline around Dronning Maud Land. The surface

temperature increase and sea ice reduction in Antarctic is minimal in this model compared to the other two, and the precipitation changes indicate change towards a drier 127k simulated Antarctic. We are unsure if the Antarctic $\delta^{18}$O extremes indicate an issue with this particular simulation. We do not consider these extreme values further here.

The H11 simulations all show larger increases in $\delta^{18}$O compared to the 127k simulations, in response to the increased SST/SAT and reduced sea ice. Here the negative precipitation seasonality changes, $\Delta\text{Precip}_{seasonal}$, are dwarfed by the

impact of the $\Delta\delta^{18}\text{O}_{seasonal}$ terms. This means that sea-ice retreat and shorter source-to-site vapour transport pathways, particularly in the colder and shoulder seasons act to strongly raise $\delta^{18}$O values (Figure 10, right column). For H11-overshoot, the Antarctic and Southern Ocean temperature and sea ice does not reverse to its pre-H11 state (*i.e.* HadCM3$_{127k}$ state) after 200 years of no meltwater: once the AMOC recovered to the initial strength (Figure A2) elevated global and polar temperature take time to cool, thus Antarctic $\delta^{18}$O values remain high more than 200 years after the H11-hosing has stopped.

Overall, for Antarctica every $\Delta\text{Precip}_{seasonal}$ result for the six simulation is small, with some negative regions (more colder season precipitation). $\Delta\delta^{18}\text{O}_{seasonal}$ is always larger and tends to dominate Antarctic $\Delta\delta^{18}$O for each simulation, mostly due to sea ice related changes (Figures 5, 9, and 10, first and last columns).

## 3.4 Implications for the interpretation of $\Delta$SAT from $\Delta\delta^{18}$O in ice cores

The interpretation of $\Delta\delta^{18}$O values in polar ice cores as a proxy for past temperatures is a critical component of paleocli-

mate reconstructions. Past LIG temperature estimates derived from $\delta^{18}$O values at key Greenland and Antarctic sites, such as NEEM, Dome C, Vostok, and Dome F, have generally been obtained by converting $\Delta\delta^{18}$O to SAT using palaeothermometer gradients. For instance in Greenland at the NEEM ice core, NEEM community members (2013) converted $\Delta\delta^{18}$O to SAT using a palaeothermometer gradient of $2.1 \pm 0.5$ K/‰, which was based on data from Vinther et al. (2009). This means that the NEEM $\delta^{18}$O value of $+3.6$ ‰ implies that, without taking account of any ice flow influences on site temperature, peak

LIG surface temperatures were $+7.5 \pm 1.8$ K warmer than the PI.

For Antarctica, Jouzel et al. (2007) applied a gradient of 1.43 K/‰, Petit et al. (1999) and Masson-Delmotte et al. (2006) used a gradient of 1.33 K/‰, whilst Sime et al. (2009), based on isotope-enabled model simulations focussed on 2100, proposed a palaeothermometer gradient for the East Antarctic Dome C (EDC) core around 1.7 K/‰, and suggested that regional and temporal variability in the paleothermometer gradient might be more significant than previously thought. The simulations

presented here enable examination of the consistency of the relationships between $\Delta$SAT and $\Delta\delta^{18}$O across all the LIG$_{127k}$ and LIG$_{H11}$ simulations.

Standard errors and uncertainties for gradients for H11 simulations are smaller then where LIG$_{127k}$ simulation results are used alone (Tables 2 and A5). This is primarily because the H11 simulations show larger $\Delta$SAT and $\Delta\delta^{18}$O values than the 127k simulations, so exert a stronger influence on the gradient of the regression lines fitted to $\Delta$SAT and $\Delta\delta^{18}$O (Figure 11 and





**Table 2.** Relationships between SAT and $\delta^{18}O$, calculated using all model simulations for each site. Fits use $\Delta\delta^{18}O$ as the dependant variable and $\Delta SAT$ as the independent variable, to enable projection of $\Delta SAT$ from $\Delta\delta^{18}O$. See also Section 2.2.3. The projected $\Delta SAT$ values are shown with 95% confidence intervals, and are based on the ice core $\Delta\delta^{18}O$ values in Table A1.

| Site | Slope | STDERR | Projected SAT |
|------|-------|--------|---------------|
| | K per ‰ | K per ‰ | K |
| Arctic core sites | | | |
| NEEM | 0.8 | 0.1 | $2.89 \pm 1.32$ |
| NGRIP | 1.08 | 0.14 | $3.36 \pm 1.92$ |
| GRIP | 1.09 | 0.17 | $3.49 \pm 2.5$ |
| GISP2 | 1.09 | 0.14 | $2.94 \pm 2.04$ |
| Camp_century | 0.73 | 0.14 | $1.46 \pm 1.78$ |
| DYE3 | 0.56 | 0.09 | $2.62 \pm 2.45$ |
| Antarctic core sites | | | |
| Vostok | 1.04 | 0.19 | $3.44 \pm 2.42$ |
| Dome F | 0.97 | 0.05 | $4.35 \pm 0.75$ |
| EDC | 1.1 | 0.1 | $4.39 \pm 1.45$ |
| EDML | 0.91 | 0.1 | $2.65 \pm 1.54$ |
| TALDICE | 0.52 | 0.2 | $1.67 \pm 3.67$ |
| Taylor | 0.62 | 0.2 | $1.18 \pm 2.67$ |

Table A1). Given Sections 3.1-3.3 show that these H11 simulations do a better job of capturing peak LIG warmth and sea ice changes at both poles, compared to the 127k simulations, this larger influence seems reasonable. We also find that if regression fits are applied to $LIG_{127k}$ simulation results alone, without including the H11 simulations, gradients are not inconsistent, if very uncertain (Table A5). This suggests that the underlying physics driving the $\delta^{18}O$-SAT relationship is consistent across different LIG climate states, even if the 127k simulations alone are insufficient to derive accurate gradients. Here we therefore

use the all simulation results to derive LIG temperature estimates from $\Delta\delta^{18}O$ values at Greenland and at Antarctic core sites.

The results in this section use primarily one-part fits (Section 2.2.3). These assume that $\Delta SAT$ and $\Delta\delta^{18}O$ are co-dependent; that they always tend to change together. However note that using a two-part fit *i.e* a regression line which permits a non-zero intercept (Table A7 and Figure A9) give results which are agree, within uncertainties, with these one-part fits. Thus this choice is not significant to our findings. Section 2.2.3 explains fit methods, and the calculation of uncertainties.

### 3.4.1 Greenland - NEEM

The interpretation of LIG temperature from $\Delta\delta^{18}O$ values measured in Greenland ice has only been attempted for the NEEM ice core (NEEM community members, 2013). This is likely because, as outlined in Section 2.3, the dating of LIG ice for most Greenland ice cores is difficult due to stratigraphic disturbances in the ice of this age. Interestingly, the simulations used here





show a substantially lower gradient of 0.8 K/‰ at NEEM, less than half of that used in NEEM community members (2013). Other ice core sites in central Greenland have a slightly higher gradient of around 1.1 K/‰ at NGRIP, GRIP, and GISP2. Based on the Section 2.3 value (in Table A1), these suggest that, without taking account of any site elevation or other ice flow effects, there was a PI to LIG warming of $+2.89 \pm 1.32$ K at NEEM, and $+3.49 \pm 2.5$ K at GRIP. Table 2 and Figures 11 and 12a show that the gradients across Greenland are relatively uniform, but with a tendency towards higher K/‰ gradients inland compared to the coast.

### 3.4.2 Antarctica

The relationship between $\Delta\delta^{18}O$ and $\Delta SAT$ shows considerable variation across Antarctica (Figure 12, Antarctic row), particularly between the high EAIS Plateau (like Vostok, Dome F and EDC) and more coastal sites (like Taldice and Taylor): gradients tend to be higher across the Plateau (Table 2). This same spatial pattern in gradients is also reflected in relationships between $\Delta\delta^{18}O$ and other quantities like $\Delta SIA$ (Figure A8). These differences underscore that it may be necessary to use site and time-specific paleothermometer gradients rather than applying a uniform and constant gradient across all Antarctic locations and periods. Like Greenland above, central Antarctica has higher K/‰ gradients compared to the coastal regions.

The gradients derived from these simulations at the Vostok, Dome F, and EDC ice core sites range from $0.9\,\mathrm{K/‰}$ to $1.1\,\mathrm{K/‰}$. These are a little lower than the canonical values often used in previous studies $(1.3 - 1.4\,\mathrm{K/‰})$ (Petit et al., 1999; Masson-Delmotte et al., 2006; Buizert et al., 2021). Applying these multi-model simulation-derived gradients to the $\Delta\delta^{18}O$ ice core values in Table A1 yields inferred past peak interglacial surface air temperature increases. These are: $4.35 \pm 0.75$ K for Dome F, $3.44 \pm 2.42$ K for Vostok, and $4.39 \pm 1.45$ K for EDC, on the High EAIS Plateau. For the more coastal sites of TALDICE and Taylor, more modest increases of $1.67 \pm 3.67$, and $1.18 \pm 2.67$ are inferred. However, if two-part fits are allowed for these coast sites *i.e* SAT can rise between PI and peak-LIG, Table A7 show larger past peak interglacial surface air temperature increases of: $2.52 \pm 1.36$, and $1.78 \pm 0.66$, which are somewhat closer to the inland core site rises.

In summary, these results together suggest a more modest PI to peak-LIG temperature increase across Antarctica compared to earlier estimates. We attribute this difference primarily to the substantially improved representation of LIG sea surface conditions around Antarctica, particularly for sea ice, which enables a better representation of the key $\Delta SAT$, $\Delta Precip$, and $\Delta\delta^{18}O$ changes.

## 4 Conclusions

The Last Interglacial (LIG), from about 130,000 to 115,000 years ago, is one of the warmest periods in recent geological history (Hoffman et al., 2017; Fischer et al., 2018), with warming that may be similar to predictions for the end of this century (IPCC, 2021). However, reconstructing peak LIG temperatures around 127 ka from ice cores can be challenging due to unknowns about the relationship between water isotopes ($\delta^{18}O$) and temperature.

This study used three isotope-enabled climate models—HadCM3, MPI-ESM-wiso, and GISS ModelE-R—in the first multi-model analysis of the LIG. Standard 127k simulations, following the PMIP4 (orbital and greenhouse gases) protocol (Otto-



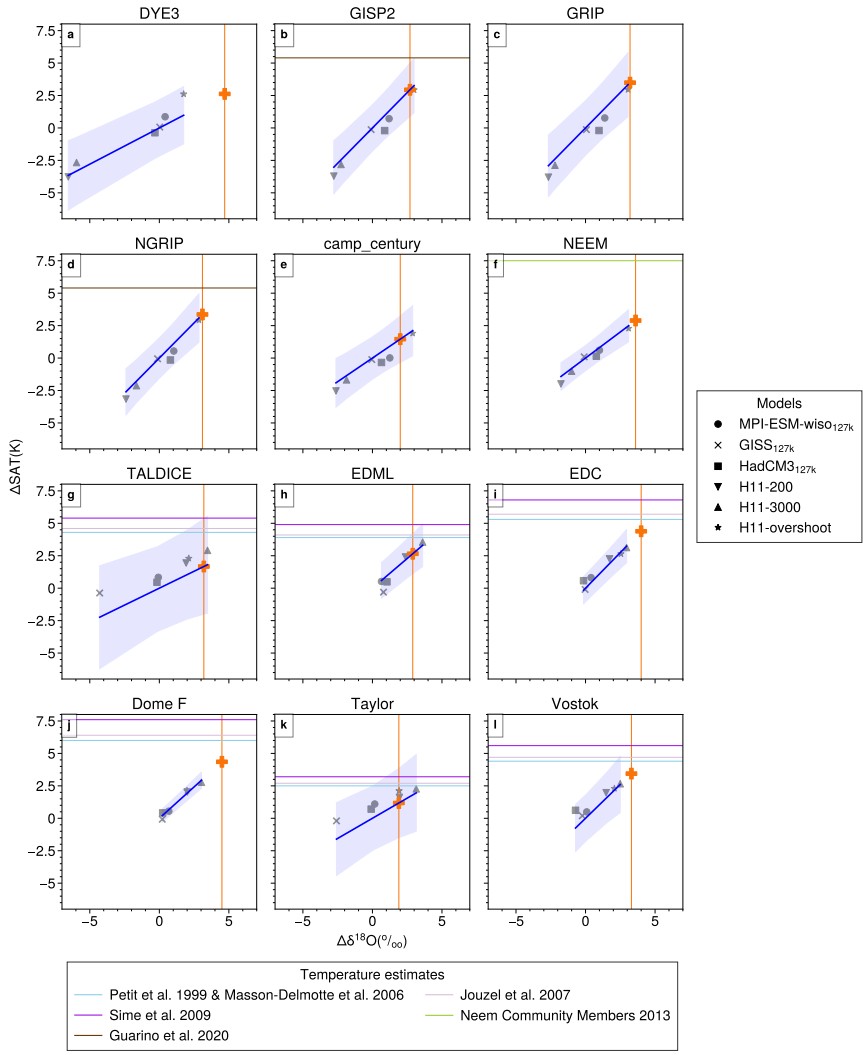

**Figure 11.** $\Delta\delta^{18}O$ versus $\Delta SAT$ for different simulations, for each ice core site. Fits are calculated as described in Section 2.2.3. Shaded bands show the 95% uncertainty envelope. Vertical orange line indicate the ice core $\Delta\delta^{18}O$ values (Table A1). Horizontal lines indicates temperature estimates for each ice core location from previous studies, whilst orange crosses indicates our estimated central $\Delta SAT$ values obtained from the simulation fits *i.e.* projected using the fitted modelled $\Delta\delta^{18}O$ - $\Delta SAT$ (blue line) relationships.



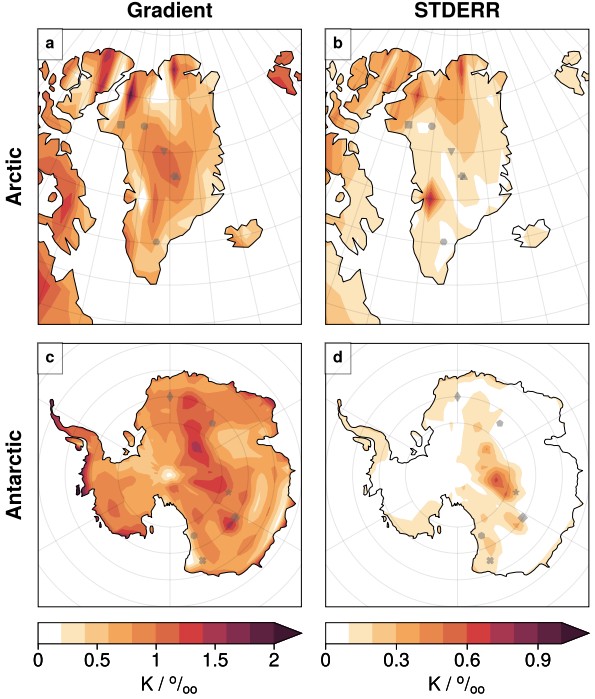

**Figure 12.** Maps of $\Delta\delta^{18}O$ versus $\Delta SAT$ gradient and associated standard error, calculated at each grid point using all simulations. Fitting procedure described in Section 2.2.3, and show for example ice core sites in Fig. 11.

Bliesner et al., 2017), were run for each model. These simulations were complemented with a long 3000-year LIG H11 simulation run using HadCM3.

The 127k simulations tend to show slight global cooling, but the H11 simulations, especially the extended H11-3000 and H11-overshoot simulations, capture significant warming trends that are consistent with the LIG observations of peak warmth

(Fischer et al., 2018; Shackleton et al., 2020). In particular, the H11-overshoot simulation appears to accurately represent peak LIG Arctic conditions, replicating 81% of the peak summer temperature increase, reducing summer sea ice by 75%, and showing periods of nearly ice-free Arctic summers. This simulation also matched the observed peak LIG changes in Greenland ice core $\delta^{18}O$, with a 2.8‰ increase, close to the observed 3.2‰. In Antarctica, the H11-3000 simulation successfully captured 50-100% of the peak LIG Southern Ocean warming and sea ice loss and 94% of the observed increases in ice core $\Delta\delta^{18}O$.

This highlights both the importance of including prolonged meltwater events in climate models to accurately capture peak LIG conditions, and the necessity of a good polar climate to simulate $\Delta SAT$, $\Delta Precip$, and $\Delta\delta^{18}O$ over the ice sheets.

Our H11 simulation is highly idealised (Holloway et al., 2018; Gao et al., 2024a), and so we emphasise it cannot be tied to particular time points within the LIG. On timing however, Shackleton et al. (2020)'s recent results suggest a LIG peak in GMOT around 129ka. Our simulation results suggest this should be co-incident with the end of the H11 - NH deglaciation - meltwater

event. This appears to in agreement with Stoll et al. (2022), who show that whilst the bulk of the Northern Hemisphere





deglacial occurred between 134-130ka, global sea level continued to increase due to meltwater until 129ka or a little after. Additionally, given H11-3000 simulates our warmest Southern Ocean and Antarctic state, and within a hundred years H11-overshoot simulates our warmest Arctic and Greenland state, peak SST and SAT temperatures, and a minimum in SIA, may have occurred within a rather short time-span for both Greenland and Antarctica. Time-scales for deeper ocean changes are however slower, and will have a different pattern of evolution. Further work should examine how short-lived the H11-overshoot is for the Arctic, alongside its interaction with the period of very-high early-summer insolation that is also centred around 127ka.

The use of $\Delta\delta^{18}O$ values in polar ice cores as a temperature proxy is key to reconstructing past climates. Temperature estimates from $\delta^{18}O$ at sites like NEEM, Dome C, Vostok, and Dome F during the LIG rely on converting $\Delta\delta^{18}O$ into $\Delta SAT$ using palaeothermometer gradients. Our simulations suggest a gradient of 0.8 K/‰ at NEEM, which is less than half that used by NEEM community members (2013), with slightly higher values, around 1.1 K/‰, at central Greenland sites such as NGRIP, GRIP, and GISP2. Without adjusting for site elevation or ice flow effects, our simulations suggest PI to LIG warming values of $+2.89 \pm 1.32$ K at NEEM and $+3.49 \pm 2.5$ K at GRIP. These are substantially lower than the previously suggested warming values. For Antarctica, our simulated $\delta^{18}O$ versus temperature gradients at Antarctic sites—Dome C, Dome F, and Vostok—range from $0.9\,K/‰$ to $1.1\,K/‰$; slightly lower than previously published values of $1.3-1.4\,K/‰$ (Petit et al., 1999; Masson-Delmotte et al., 2006). Applying these updated gradients, the estimated peak temperatures increases between the PI and LIG are $4.39 \pm 1.45\,K$ for Dome C, $4.35 \pm 0.75\,K$ for Dome F, and $3.44 \pm 2.42\,K$ for Vostok.

These findings suggest a more modest temperature rise during the LIG for both Greenland and Antarctica than previously thought. This revision is largely due to better representation of polar sea ice changes in the new extended H11 simulations. However, it is emphasised that these revised temperature estimates do not account for any ice core site elevation or other ice flow-related impacts on site temperatures. In the future, efforts should firstly continue to better characterize LIG Antarctic and Greenland ice sheet changes, which will have an influence on $\delta^{18}O$ and temperature (Holloway et al., 2016; Werner et al., 2018; Domingo et al., 2020; Goursaud et al., 2021; Zou et al., 2025). And secondly, a re-examination of the nature of peak polar ice sheet warmth and its co-incidence in Antarctica and Greenland is required.

*Code availability.* All code used to produce the figures and tables is available on zenodo: DOI:10.5281/zenodo.14640496.

*Data availability.* The model datasets used to produce the figures and tables are available on zenodo: DOI:10.5281/zenodo.14640496. Last Interglacial summer air temperature observations for the Arctic (Version 1.0) are also published by the NERC EDS UK Polar Data Centre here: https://doi.org/10.5285/9AB58D27-596A-472C-A13E-2DCD68612082. The Capron et al. (2017) and Hoffman et al. (2017) datasets are available from: https://doi.org/10.1016/j. quascirev.2017.04.019. The Chandler and Langebroek (2021a) dataset is available from: https://doi.org/10.1594/PANGAEA.938620. The Chadwick et al. (2021) dataset is available from: https://doi.org/10.1594/PANGAEA.936573. The four paleoclimate syntheses by Gao et al. (2024) from: https://zenodo.org/records/11079974. The Greenland ice core data is available in Table 1 of: https://doi.org/10.1029/2019JF005237. The Antarctic ice core data is available in Table S3 of: https://doi.org/10.1029/2020GL091412.



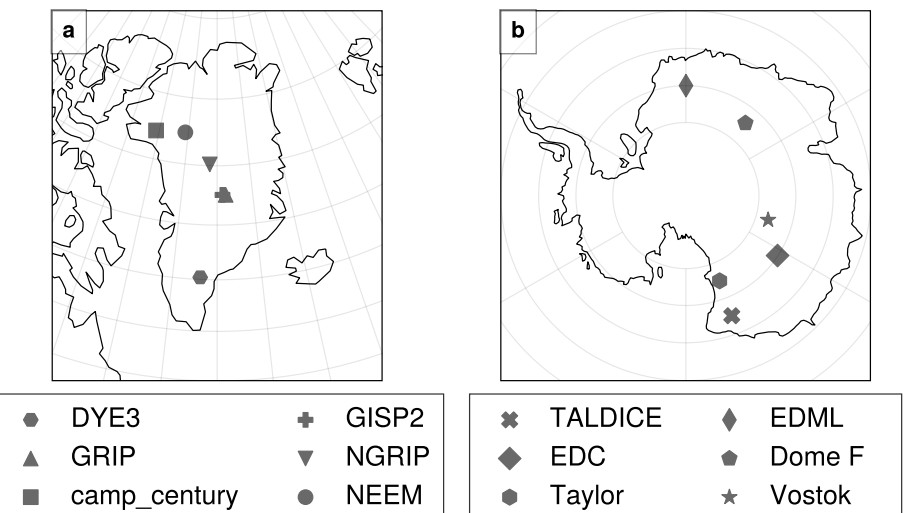

**Figure A1.** The locations of the Greenland and Antarctic ice core sites.

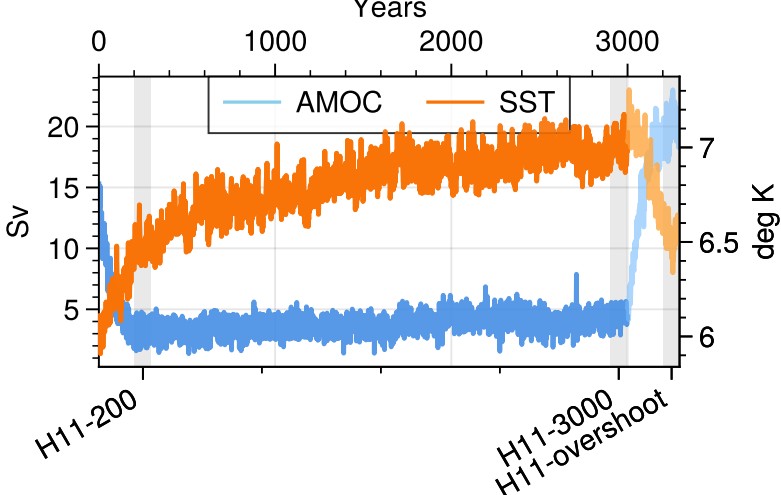

**Figure A2.** Time series of AMOC (in Sv) and average Southern Hemisphere SST (south of 40°S) time series from HadCM3 H11 simulations. The lighter colors represent the period where the water hosing was stopped. The three grey bands denote the three selected slices from this simulation: H11-200; H11-3000; and H11-overshoot.

**Appendix A**

**A1**





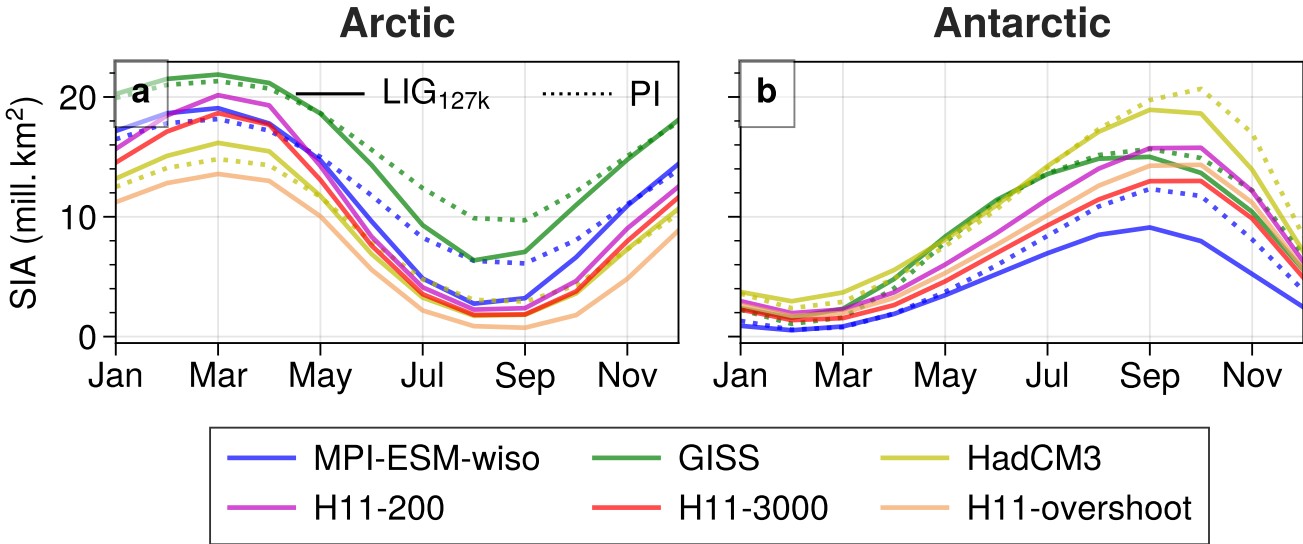

**Figure A3.** Seasonal cycle of total sea ice area in the Northern and Southern Hemisphere, for each simulation. These are absolute values (not Δ values). LIG values are shown using solid lines and PI values are the dotted lines.



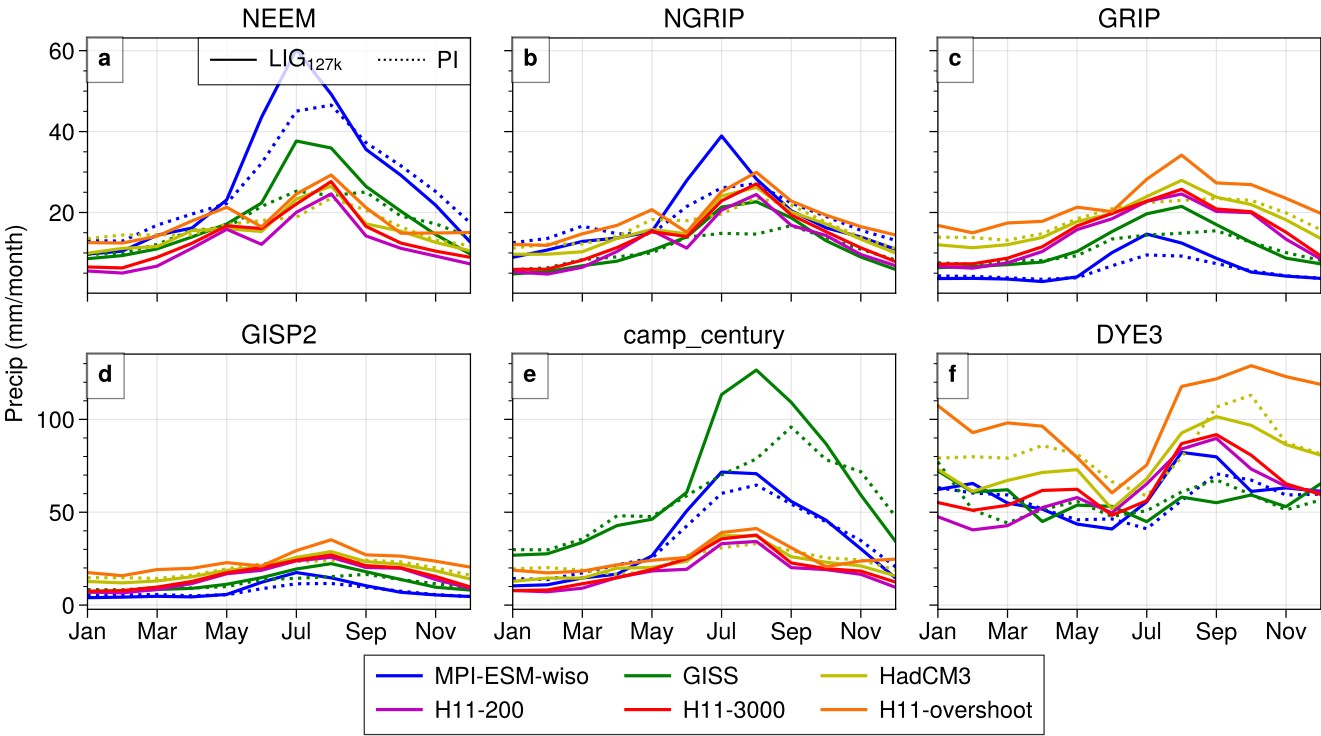

**Figure A4.** Seasonal cycle of precipitation at each ice core site in Greenland. These are absolute values (not Δ values). LIG values are shown using solid lines and PI values are the dotted lines.





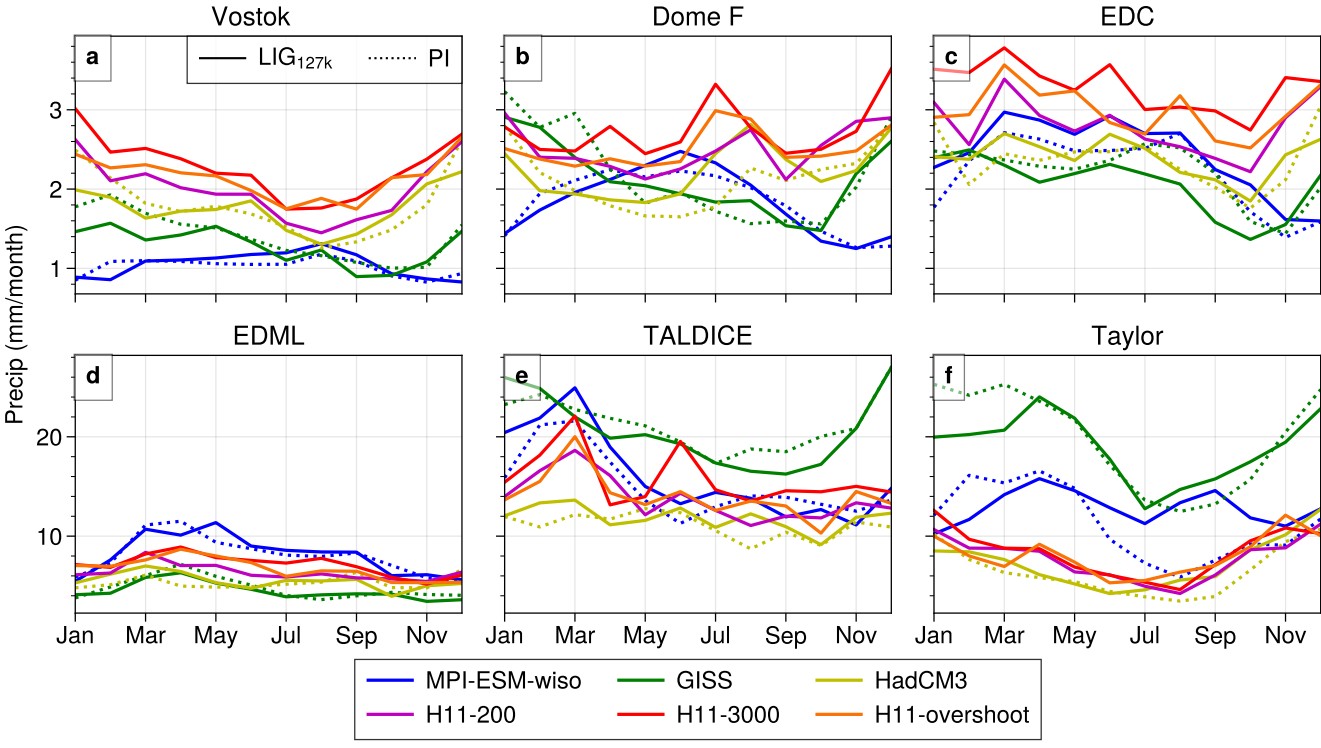

**Figure A5.** The seasonal cycle of precipitation, as Figure A4, but for Antarctica.





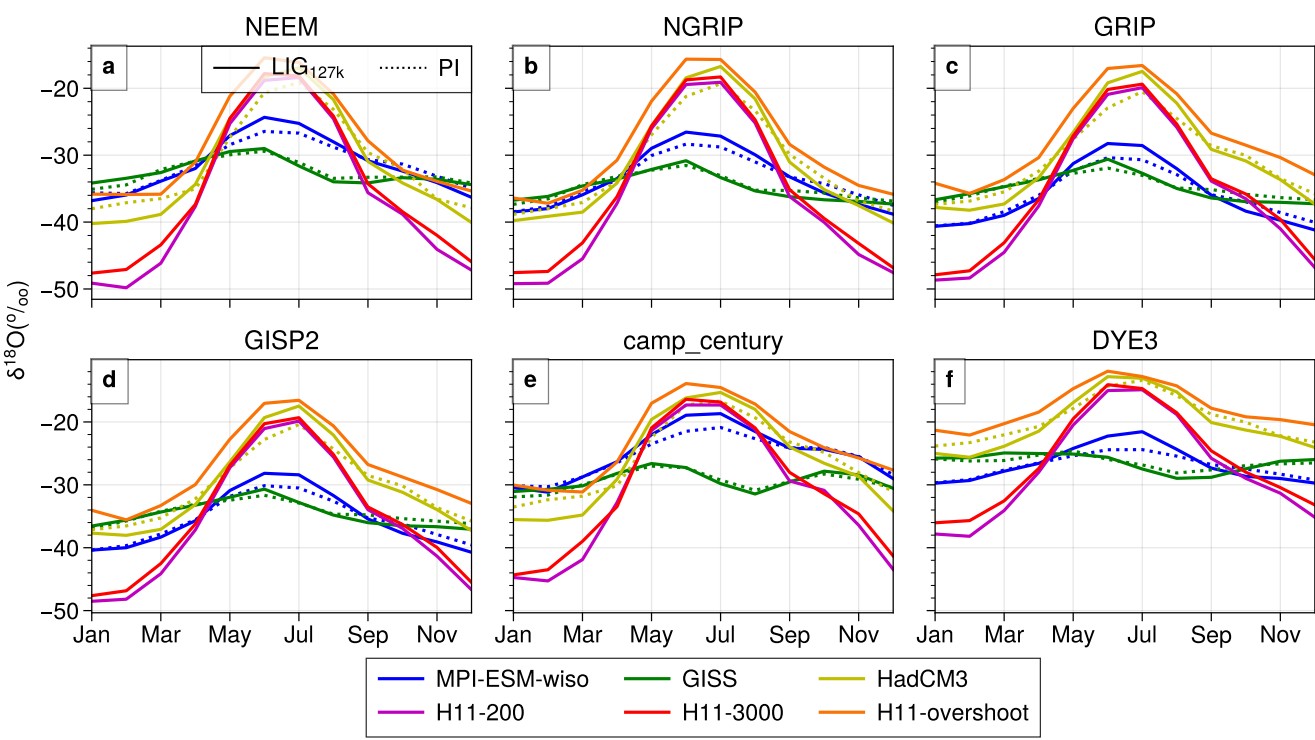

**Figure A6.** The seasonal cycle of $\delta^{18}O$, otherwise as Figure A4.





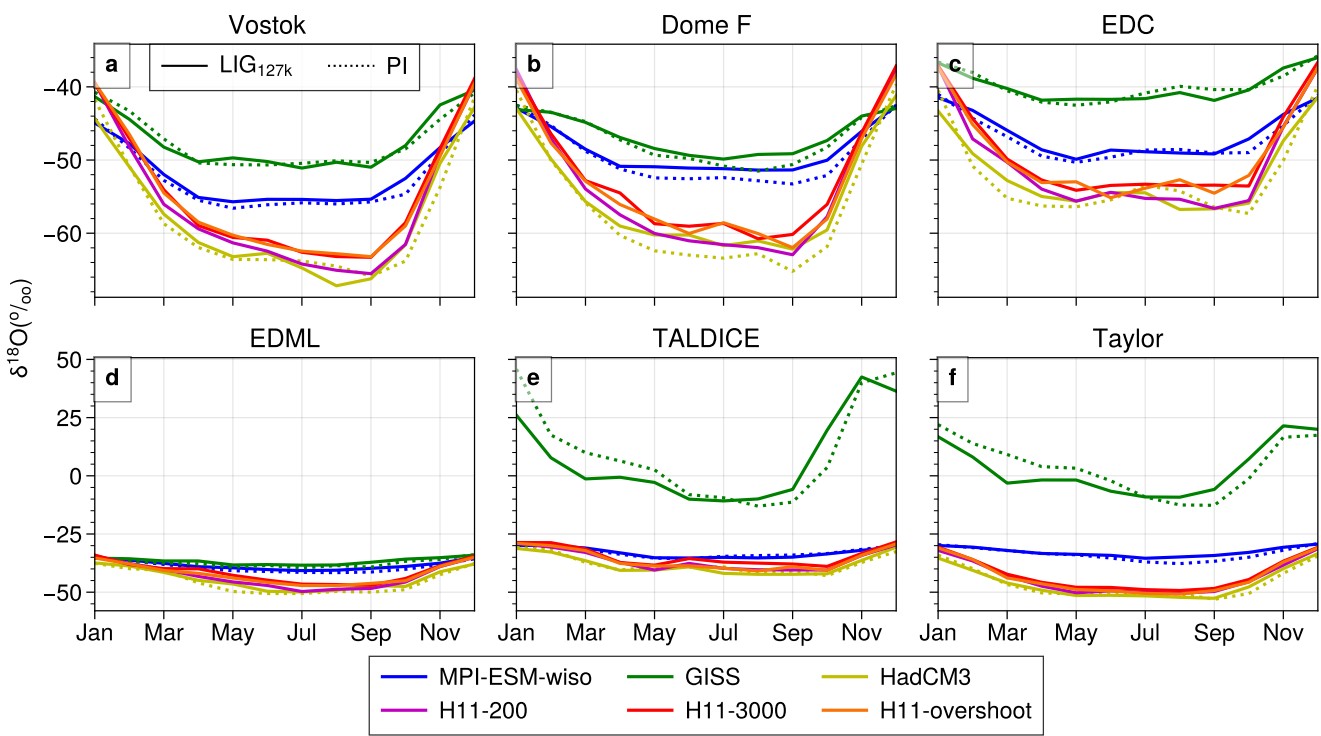

**Figure A7.** The seasonal cycle of δ¹⁸O for Antarctica, otherwise as Figure A6.




**Figure A8.** $\Delta$SAT at each ice core location (first column), $\Delta$ SST (north/south of 40°latitude for Arctic and the Antarctica) and $\Delta$SIA (annual sea ice area) versus $\Delta\delta^{18}O$ from each ice core sites, for all simulations.




**Figure A9.** Same as 11, but using a two-part fit *i.e* a regression line which permits a non-zero intercept (orange lines).





**Table A1.** Ice core and simulated $\Delta\delta^{18}O$ values at ice core locations, and simulated $\Delta$SAT.

| SITE | LON | LAT | Ice Core $\Delta\delta^{18}O$ | MPI-ESM-wiso$_{127k}$ $\Delta$SAT | MPI-ESM-wiso$_{127k}$ $\Delta\delta^{18}O$ | GISS$_{127k}$ $\Delta$SAT | GISS$_{127k}$ $\Delta\delta^{18}O$ | HadCM3$_{127k}$ $\Delta$SAT | HadCM3$_{127k}$ $\Delta\delta^{18}O$ | H11-200 $\Delta$SAT | H11-200 $\Delta\delta^{18}O$ | H11-3000 $\Delta$SAT | H11-3000 $\Delta\delta^{18}O$ | H11-overshoot $\Delta$SAT | H11-overshoot $\Delta\delta^{18}O$ |
|---|---|---|---|---|---|---|---|---|---|---|---|---|---|---|---|
| **Greenland Core Sites** | | | | | | | | | | | | | | | |
| NEEM | -51.1 | 77.4 | 3.6 | 0.6 | 1.0 | 0.1 | -0.1 | 0.1 | 0.8 | -2.0 | -1.8 | -1.3 | -1.0 | 2.3 | 3.1 |
| NGRIP | -42.3 | 75.1 | 3.1 | 0.5 | 1.0 | -0.0 | -0.1 | -0.2 | 0.8 | -3.2 | -2.4 | -2.3 | -1.6 | 2.9 | 2.8 |
| GRIP | -37.6 | 72.6 | 3.2 | 0.8 | 1.4 | -0.1 | 0.1 | -0.2 | 1.0 | -3.8 | -2.7 | -3.0 | -2.2 | 3.0 | 3.1 |
| GISP2 | -38.5 | 72.6 | 2.7 | 0.7 | 1.2 | -0.1 | -0.1 | -0.2 | 0.9 | -3.7 | -2.8 | -2.9 | -2.3 | 2.9 | 3.0 |
| Camp Century | -61.1 | 77.2 | 2.0 | 0.0 | 1.3 | -0.1 | -0.0 | -0.3 | 0.6 | -2.5 | -2.6 | -2.0 | -1.9 | 1.9 | 2.9 |
| DYE3 | -43.8 | 65.2 | 4.7 | 0.9 | 0.4 | 0.1 | 0.0 | -0.4 | -0.3 | -3.8 | -6.6 | -2.9 | -6.0 | 2.6 | 1.7 |
| **Mean** | - | - | 3.22 | 0.6 | 1.0 | -0.0 | -0.0 | -0.2 | 0.6 | -3.2 | -3.1 | -2.4 | -2.5 | 2.6 | 2.8 |
| **Antarctic Core Sites** | | | | | | | | | | | | | | | |
| Vostok | 106.5 | -78.3 | 3.3 | 0.5 | 0.1 | 0.2 | -0.2 | 0.6 | -0.7 | 2.0 | 1.5 | 2.7 | 2.5 | 2.3 | 2.1 |
| Dome F | 39.4 | -77.2 | 4.5 | 0.6 | 0.7 | -0.1 | 0.2 | 0.4 | 0.2 | 2.0 | 2.0 | 3.0 | 3.0 | 2.1 | 2.0 |
| EDC | 123.2 | -75.1 | 4.0 | 0.8 | 0.4 | -0.1 | -0.0 | 0.6 | -0.1 | 2.3 | 1.7 | 3.4 | 3.0 | 2.6 | 2.5 |
| EDML | 0.0 | -75.0 | 2.9 | 0.5 | 0.6 | -0.3 | 0.8 | 0.5 | 1.0 | 2.4 | 2.4 | 3.7 | 3.6 | 2.8 | 3.0 |
| TALDICE | 159.1 | -72.5 | 3.2 | 0.8 | -0.1 | -0.4 | -4.3 | 0.4 | -0.2 | 1.9 | 1.9 | 3.0 | 3.5 | 2.3 | 2.1 |
| Taylor | 158.4 | -77.5 | 1.9 | 1.1 | 0.2 | -0.2 | -2.6 | 0.7 | -0.1 | 1.6 | 1.9 | 2.3 | 3.2 | 2.1 | 1.9 |
| **Mean** | - | - | 3.3 | 0.7 | 0.3 | -0.1 | -1.0 | 0.5 | 0.0 | 2.0 | 1.9 | 3.0 | 3.1 | 2.4 | 2.3 |

**Table A2.** Global mean $\Delta$SAT, $\Delta$ SST, and $\Delta$ GMOT (Global Mean Ocean Temperature) for each simulation.

| Experiment | $\Delta$SST | $\Delta$SAT | $\Delta$GMOT |
|---|---|---|---|
| MPI-ESM-wiso$_{127k}$ | -0.07 | -0.11 | -0.045 |
| GISS$_{127k}$ | -0.17 | -0.15 | - |
| HadCM3$_{127k}$ | -0.14 | -0.23 | -0.016 |
| H11-200 | 0.04 | -0.23 | - |
| H11-3000 | 0.45 | 0.35 | 0.84 |
| H11-overshoot | 0.83 | 1.09 | 0.78 |



**Table A3.** Simulated Arctic and Antarctic ΔSAT and ΔSIA values. Column 1: mean Δ SAT (averaged north of 60°N for Arctic and south of 60°S for Antarctic). Column 2: annual LIG sea ice area (mill. km$^2$). Column 3: LIG-PI anomaly in SIA (in %). Column 4: summer mean Δ SAT (JJA averaged north of 60°N for Arctic and DJF south of 60°S for Antarctica). Column 5: seasonal minimum LIG sea ice area (mill. km$^2$). Column 6: LIG-PI anomaly in seasonal minimum SIA (in %).

| Simulation | Annual ΔSAT | Annual SIA (mill km$^2$) | Annual ΔSIA (%) | Summer ΔSAT | Monthly minimum SIA (mill km$^2$) | Monthly minimum ΔSIA (%) |
|---|---|---|---|---|---|---|
| | | | Arctic | | | |
| MPI-ESM-wiso$_{127k}$ | 0.4 | 11.6 | -6.9 | 1.8 | 2.7 | -55. 0 |
| GISS$_{127k}$ | 0.91 | 15.4 | -5.1 | 2.69 | 6.4 | -34.5 |
| HadCM3$_{127k}$ | -0.32 | 8.9 | -0.8 | 1.67 | 1.7 | -40.8 |
| H11-200 | -2.68 | 10.9 | 21.5 | 0.82 | 2.3 | -22.9 |
| H11-3000 | -1.51 | 9.9 | 10.5 | 1.23 | 1.8 | -38. 0 |
| H11-overshoot | 2.18 | 7.1 | -20.7 | 3.02 | 0.7 | -74.7 |
| | | | Antarctic | | | |
| MPI-ESM-wiso$_{127k}$ | 0.84 | 4.4 | -23.7 | -0.77 | 0.5 | -6.8 |
| GISS$_{127k}$ | 0.01 | 8.7 | -2.4 | -0.02 | 1.6 | 48.4 |
| HadCM3$_{127k}$ | 0.45 | 10.4 | -3.2 | -1.64 | 3. 0 | 24.5 |
| H11-200 | 1.95 | 8.4 | -21.8 | -5.58 | 2. 0 | -17.3 |
| H11-3000 | 3.12 | 6.7 | -37.3 | -3.76 | 1.4 | -42.1 |
| H11-overshoot | 2.52 | 7.6 | -29.7 | 1.79 | 1.7 | -27.4 |



**Table A4.** Simulated Southern Ocean SST and sea ice values, and comparison with equivalent observational syntheses from Gao et al. (2024a). Left-most columns show average annual and summer ΔSST, for the Southen Ocean (south of 40°S) for each simulation. Right columns show SST synthesis averages (top row) as compiled by Gao et al. (2024a). Lower rows show the percentage of the observed core-site ΔSST values achieved by each simulation, for each data synthesis. See text and Gao et al. (2024a) for details of data synthesis.

| | | | EC2017 | | JH2017 | | DC2021 | | MC2021 | |
|---|---|---|---|---|---|---|---|---|---|---|
| | Annual | Summer | Annual | Summer | Annual | Summer | Annual | Summer | Summer | September |
| | ΔSST | ΔSST | ΔSST | ΔSST | ΔSST | ΔSST | ΔSST | ΔSST | ΔSST | ΔSIA (%) |
| Synthesis mean | - | - | 4.0K | 1.7K | 2.6K | 1.7K | 2.2K | 2.2K | 1.2K | -41.3 |
| | | | % of change captured by each simulation | | | | | | | |
| MPI-ESM-wiso$_{127k}$ | 0.26 | -0.11 | 6.4 | -6.1 | 10.2 | -6.4 | 12.1 | -4.8 | -8.5 | 63.3 |
| GISS$_{127k}$ | -0.03 | -0.38 | -0.7 | -21.8 | -1.1 | -22.9 | -1.3 | -17.1 | -30.7 | 10.2 |
| HadCM3$_{127k}$ | 0.14 | -0.12 | 3.5 | -6.7 | 5.6 | -7.1 | 6.6 | -5.3 | -9.5 | 9.9 |
| H11-200 | 0.89 | 0.76 | 21.9 | 43.7 | 34.6 | 46.0 | 41.0 | 34.4 | 61.6 | 49.2 |
| H11-3000 | 1.38 | 1.28 | 34.2 | 73.5 | 53.9 | 77.2 | 63.9 | 57.7 | 103.4 | 83.2 |
| H11-overshoot | 0.95 | 0.87 | 23.6 | 49.6 | 37.2 | 52.1 | 44.2 | 39.0 | 69.8 | 67.4 |

**Table A5.** Fits and projected ΔSAT values based on ice core Δδ$^{18}$O, as Table 2, but based solely on the LIG$_{127k}$ simulations (H11 results excluded).

| Site | Slope | STDERR | Projected SAT |
|---|---|---|---|
| | K per ‰ | K per ‰ | K |
| Arctic core sites | | | |
| NEEM | 0.44 | 0.16 | $1.58 \pm 2.65$ |
| NGRIP | 0.26 | 0.24 | $0.81 \pm 3.5$ |
| GRIP | 0.30 | 0.26 | $0.97 \pm 4.04$ |
| GISP2 | 0.31 | 0.28 | $0.84 \pm 3.76$ |
| Camp_century | -0.10 | 0.17 | $-0.2 \pm 1.74$ |
| DYE3 | 1.74 | 0.31 | $8.19 \pm 6.37$ |
| Antarctic core sites | | | |
| Vostok | -0.77 | 0.53 | $-2.54 \pm 7.67$ |
| Dome F | 0.81 | 0.31 | $3.63 \pm 6.11$ |
| EDC | 1.36 | 1.36 | $5.42 \pm 23.61$ |
| EDML | 0.28 | 0.32 | $0.82 \pm 4.43$ |
| TALDICE | 0.08 | 0.16 | $0.25 \pm 3.65$ |
| Taylor | 0.09 | 0.35 | $0.18 \pm 4.91$ |



**Table A6.** Fits and projected $\Delta$SAT values based on ice core $\Delta\delta^{18}$O, as Table 2, but based solely on the H11 simulations (LIG$_{127k}$ results excluded).

| Site | Slope | STDERR | Projected SAT |
|------|-------|--------|---------------|
|      | K per ‰ | K per ‰ | K |
| Arctic core sites | | | |
| NEEM | 0.85 | 0.12 | $3.05 \pm 2.65$ |
| NGRIP | 1.17 | 0.10 | $3.63 \pm 2.22$ |
| GRIP | 1.20 | 0.15 | $3.83 \pm 3.56$ |
| GISP2 | 1.17 | 0.11 | $3.16 \pm 2.66$ |
| Camp_century | 0.82 | 0.10 | $1.63 \pm 2.09$ |
| DYE3 | 0.55 | 0.14 | $2.61 \pm 6.06$ |
| Antarctic core sites | | | |
| Vostok | 1.13 | 0.07 | $3.72 \pm 1.41$ |
| Dome F | 0.97 | 0.05 | $4.38 \pm 1.21$ |
| EDC | 1.09 | 0.07 | $4.38 \pm 1.69$ |
| EDML | 0.96 | 0.03 | $2.79 \pm 0.77$ |
| TALDICE | 0.93 | 0.07 | $2.96 \pm 1.77$ |
| Taylor | 0.82 | 0.11 | $1.57 \pm 2.12$ |





**Table A7.** This table is as 2, but using a two-part regression, *i.e.* allowing non-zero intercepts. Both the gradient and the intercept are used in the calculation of the projected temperature.

| Site | Slope | Intercept | STDERR | Projected SAT |
|------|-------|-----------|--------|---------------|
|  | K per ‰ | K | K per ‰ | K |
| *Arctic core sites* | | | | |
| NGRIP | 1.1 | -0.43 | 0.1 | 2.97 ± 1.51 |
| GRIP | 1.11 | -0.66 | 0.09 | 2.88 ± 1.48 |
| GISP2 | 1.09 | -0.52 | 0.09 | 2.41 ± 1.46 |
| Camp_century | 0.73 | -0.5 | 0.08 | 0.98 ± 1.21 |
| DYE3 | 0.63 | 0.58 | 0.09 | 3.56 ± 2.67 |
| *Antarctic core sites* | | | | |
| Vostok | 0.77 | 0.71 | 0.11 | 3.24 ± 1.22 |
| Dome F | 0.98 | -0.03 | 0.09 | 4.39 ± 1.00 |
| EDC | 0.94 | 0.4 | 0.1 0 | 4.14 ± 1.23 |
| EDML | 1.19 | -0.71 | 0.14 | 2.73 ± 1.22 |
| TALDICE | 0.43 | 1.13 | 0.07 | 2.52 ± 1.36 |
| Taylor | 0.45 | 0.93 | 0.05 | 1.78 ± 0.68 |





*Author contributions.*   LCS led the development of this study. RS, MW, AC, and ANL ran simulations. RS performed all data analysis. LCS
wrote the first draft of this manuscript. All authors contributed to the final draft.

*Competing interests.*   Some authors are members of the editorial board of Climate of the Past.

*Acknowledgements.*   LCS is supported in this work through: Past-to-Future: Towards fully paleo-informed future climate projections (P2F),
GN 101184070; The Sensitivity of the West Antarctic Ice Sheet to +2C: SWAIS2C, NE/X009386/1. Assessing ocean-forced, marine-
terminating glacier change in Greenland during climatic warm periods and its impact on marine productivity: KANG-GLAC, NE/V006509/1.
EMcC, LCS, and RS are supported through European Research Council H2020 Grant Number 864637 (ANTarctic Sea Ice Evolution from a
novel biological archive, ANTSIE). AdB is supported through Swedish Research Council grant VR 2020-04791.ANL thanks NASA High-
End Computing Program for computing resources through the NASA Center for Climate Simulation at Goddard Space Flight Center and
NASA GISS for institutional support, particularly interns of the NASA Climate Change Research Initiative program. AC and MW were
supported by the German Federal Ministry of Education and Research (BMBF) as a Research for Sustainability initiative (FONA). MPI-
ESM-wiso simulation was performed at the German Climate Computing Center (DKRZ).



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
