# Peer review of "H11 meltwater and standard 127ka Last Interglacial simulations suggest more modest peak temperatures for both Greenland and Antarctica: A multi-model study of water isotopes"

_EGUsphere, 2025_

## Author Comment (AC1)

In response to these thoughtful and expert reviews, we have made a series of minor and more major revisions to the manuscript, to make the improvements the referees request:

**(1) We now cite Liu et al. (2023) and have incorporated new discussion material addressing their findings**. In particular, we acknowledge the potential role of Antarctic inversion layers in altering the (T–$\delta^{18}$O) slope and highlight the importance of this issue in the interpretation of ice core records both in the Introduction and Conclusions.

**(2) We have substantially rewritten the latter part of the Introduction, so that it now covers a much broader range on literature about controls on D18O.** This follows the very helpful guidance provided by Reviewer 2.

**(3) We have retitled the manuscript to better reflect the importance of the H11 event**, whilst also retaining a focus on the important – and novel - multi-model aspect of the study too.

**(4) We have expanded our discussion of LGM climate and isotopic model–data comparisons in Section 3.4**, drawing on existing literature to place our LIG findings in a broader paleoclimate context. This helps clarify whether the model–data mismatch observed during the LIG reflects general model limitations or is more likely due to missing forcings specific to the LIG, such as Heinrich event impacts.

**(5) Conclusions have been substantially modified** to include comments and all new references on vegetation and ice sheet feedbacks, alongside adding a new final concluding point on how further progress could be made on better evaluating controls on d18O in future.

**(6) Around 15 new references** have been added during the revisions, including all of those suggested by the referees.

Changes to the text denoted in ***bold italics.***

**RC1**: 'Comment on egusphere-2025-288', Anonymous Referee #1, 02 Feb 2025
This paper studies the climate/d18O response at the LIG with the focus on the Arctic and Antarctic regions. The authors used 4 isotope-enabled climate models under 127 climate forcing, and one model with a long hosing simulating the H11 event. The major conclusion seems to be the model response is too small relative to the ice core observations in both temperatura and d18O, except perhaps the H3000-overshoot. It seems to me this paper has two major points. First, there is a systematic model-data inconsistency, with the model of less signal than in observation. Second, the H11 events indeed tends to reduce the model-data inconsistency, and therefore may be an important factor in the real world LIG resposne. It is a useful paper that summrizes the current state-of-the-art modeling of climate/d18O on LIG. Nevertheless, I think the paper can be further improved before publications.

**Major concern:**
Comparison with LGM: The LIG model-data comparsion will be better compared in the context of LGM model-data comparison of the same models. (I assume these models have done the LGM experiments before LIG experiments). Are all models has the similar inconsistency with observations? This LGM comparison has two advantages. First, observational data should have more uncertainty at the LIG than at LGM, while the model uncertainty is the same at LIG and LGM. Second, LGM clearly has no influence of H events (because it is well separated from H1 and H2). So, if all the models also have less signals in temperature/d18O at LGM than in observations, it is more likely that the model-data inconsistency is caused by the model deficiency. Otherwise, H11 may be a more important factor in reconciling the model-data discrepancy.

Thank you for this very interesting and constructive suggestion. Unfortunately, we do not have equivalent LGM simulations available for all the models used in our study. As a result this, and also to a degree space and time limitations, a systematic LGM–LIG comparison using the same set of models is not currently feasible. We also note that it is not entirely clear that observational constraints are more certain for the LGM than for the LIG. In particular, LGM boundary conditions—especially the shape and extent of the Antarctic and Greenland ice sheets—are subject to considerable uncertainty, which may affect model–data comparisons. That said, our primary motivation for focusing on the LIG is that isotope-enabled simulations are highly sensitive to the simulated climate, and we are specifically interested in improving the interpretation of LIG ice core records. Given this goal, we believe it is most appropriate to target the LIG directly. Nonetheless, we fully appreciate the importance of placing our LIG results in a broader paleoclimate context. In response to your suggestion, we added a short discussion of how these results compare to LGM climate and isotopic model–data results in Section 3.4. This draws on existing literature, and helps frame the LIG findings and clarify whether the model–data mismatch may reflect more general model limitations or forcings specific to the LIG.

New Section 3.4 paragraph: *The gradients derived from these simulations at the Vostok, Dome F, and EDC ice core sites range from 0.9 K/‰ to 1.1 K/‰. These are a little lower than the canonical values often used in previous studies (1.3 − 1.4 K/‰) (Petit et al., 1999; Masson-Delmotte et al., 2006; Buizert et al., 2021).* ***For example, Werner et al. (2018) set up eight atmospheric model simulations using ECHAM5-wiso with different Last Glacial Maximum (LGM) ice sheet reconstructions to investigate the temporal and spatial relationships between δ18O and SAT across Antarctica. They found that, whilst these LGM to PI relationships differed across Antarctic regions, they tended to be close to a commonly identified relationship of around 1.25 K/‰. That said, Cauquoin et al. (2023) note that ECHAM5-wiso tends to significantly overestimate isotope changes in Antarctica compared to more recent ECHAM6-wiso simulations, even under identical boundary conditions. Thus some questions do remain LGM to PI △δ18O and △SAT relationships. Indeed it is possible, as yet, we do not have reliable multi-model isotopically enable LGM simulations.***

Minor concerns:

•       The strategy to use cross-model T-d18O slope is no guarantee the model slope is more correct than real world. Even if the slope is the same as in real world, the small signal in both temperatura and d18O are consistent with the data-model inconsistency anyway. It is an interesting try, but does not give much information.

Thank you for this thoughtful observation. We fully agree that there is no guarantee the real-world PI-to-LIG temperature–$\delta^{18}O$ (T–$\delta^{18}O$) slope matches the cross-model ensemble slope. However, our intention was to explore a plausible framework for interpreting model–data discrepancies by examining relationships within and between models. While we recognize the limitations of this approach, we hope the reviewer appreciates that our aim is to provide a structured way to assess how various forcings affect both temperature and $\delta^{18}O$, even if the absolute slope remains uncertain.

•       The major result is model-data inconsistency, not just in the Arctic and Antarctic, but also for the global mean temperatura. The paper title, however, implies a bias in observation.

This is correct; the findings do indeed depend on whether the LIG climate is accurately simulated. However, as the reviewer notes, the focus of the manuscript is essentially on the

PI-to-LIG temperature and δ¹⁸O changes, the associated (T–δ¹⁸O) slope, and the implications for interpreting past temperatures from PI-to-LIG ice core δ¹⁸O changes. Previously, PI-to-LIG δ¹⁸O changes in ice cores were often interpreted as indicating a very large temperature increase. This paper shows that such an interpretation is likely incorrect— hence, a bias in the inferred temperature from PI-to-LIG ice core δ¹⁸O observations.

• The paper should address the inversion layer problem more explicitly. This is potentially serious in the Antactica and has been shown recently to disrupt the T-d18O slope dramatically, at least, at LGM (Liu et al. (2023).

This is a very good point. The study by Liu et al. (2023) provides important insight into how changes in inversion layer temperatures can strongly affect the (T–δ¹⁸O) slope in Antarctica, particularly at the LGM. Unfortunately, we do not have access to the necessary model output to directly assess inversion layer temperature changes in our simulations. However, in response to this comment, we have now cited Liu et al. (2023) in the manuscript and added new discussion material that reflects their findings. We also highlight in the conclusions that further work is needed to understand the role of inversion layers in shaping δ¹⁸O– temperature relationships during the LIG. See also below for rewritten sections.

Liu, Z. et al., 2023: Reconstructing past Antarctic temperature using present seasonal d18O-inversion layer temperature: Unified Slope Equations and application. J. Clim., 36, 2933-2957,
* * *
RC2: 'Reviewer comment on egusphere-2025-288', Jesper Sjolte, 21 Feb 2025
**Summary of Sime et al.**
Sime et al. use an isotope enabled model ensemble to estimate the δ18O-temperature slope during the Last Interglacial (LIG). The authors explore model decencies, impacts of the seasonality of precipitation and sea ice anomalies. Apart from the PMIP protocol LIG simulations, Sime et al. explore a H11 overshoot scenario, which, unlike the PMIP simulations, explains most of the amplitude of LIG ice core isotope anomalies in both Greenland and Antarctica. Combining all simulations the conclusion is an estimated LIG temperature anomaly with about half the amplitude for Greenland compared to previous studies.

**General comments.**
I find that the overshoot scenario part in this study to be the most interesting and novel aspect, and I wish the authors would focus more, if not entirely on this topic. The standard simulations presented in this work mainly reflect the discrepancies between simulations and ice core data during LIG found by previous studies. The rationale behind the overshoot scenario could be explained more in depth, expanding on the section L56-66. While the model setup and the use of multiple models could be tested more in future studies, I think the overshoot scenario is a viable explanation for a large part (80%?) of the discrepancy between models and ice core data, unless there are other reasons that the authors do not consider this an option? The remaining part of the simulated isotope signal could be related to vegetation feedbacks (Claussen et al., 2006; Schugers et al., 2007; Nikolova et al., 2013), and uncertainties in ice sheet configuration (e.g., Otto-Bliesner et al., 2006).

We agree that the overshoot scenario is one of the most interesting and novel aspects of this study. However, we also wish to emphasize that this is the first-ever multi-model isotope-enabled study of the LIG, and as such, it provides an important baseline by showing that standard 127 ka simulations consistently fail to match observations across all models.

Establishing this multi-model consistency in the model–data mismatch is, in our view, a key contribution in itself. Regarding the overshoot experiment, the H11-overshoot simulation produces a Greenland core-mean $\Delta\delta^{18}O$ increase of 2.8‰, compared to an observed core-mean $\Delta\delta^{18}O$ of about 3.2‰. In terms of both Arctic summer $\Delta$SAT and Greenland $\delta^{18}O$, this implies that the H11-overshoot accounts for roughly 85% of the observed signal. We agree this is a substantial improvement and highlights the potential importance of meltwater events like H11 in shaping the LIG climate signal. As the reviewer suggests, the remaining (~15%) model–data discrepancy may partly be explained by vegetation feedbacks (e.g., Claussen et al., 2006; Schurgers et al., 2007; Nikolova et al., 2013) or uncertainties in ice sheet configuration. While we currently do not have simulations available to test the vegetation feedback hypothesis directly, we hope that future experiments, for example within the PMIP7 interglacials project, will explore these mechanisms in greater depth. We have added a comment in the conclusions to reflect this.

The conclusions are modified to reflect this interest in two places, here: '*In particular, the H11-overshoot simulation appears to accurately represent peak LIG Arctic conditions, replicating 81% of the peak summer temperature increase, reducing summer sea ice by 75%, and showing periods of nearly ice-free Arctic summers. This simulation also matched the observed peak LIG changes in Greenland ice core δ18O, with a 2.8 ‰ increase, close to the observed 3.2 ‰. **We note that some parts of the remaining 0.4‰ missing part of the simulated isotope signal could be due to vegetation or ice sheet feedbacks which are not included in these simulations (Claussen et al., 2006; Schurgers et al., 2007; Nikolova et al., 2013; Otto-Bliesner et al., 2006).*'**

And also here: '*However, it is emphasised that these revised temperature estimates do not account for any ice core site elevation or other ice flow-related impacts on site temperatures. In the future, efforts should firstly continue to better characterize LIG Antarctic and Greenland ice sheet changes, which will have an influence on δ18O and temperature (Holloway et al., 2016; Werner et al., 2018; Domingo et al., 2020; Goursaud et al., 2021; **Zou et al., 2025), alongside possible vegetation feedbacks (Claussen et al., 2006; Schurgers et al., 2007; Nikolova et al., 2013; Otto-Bliesner et al., 2006).*'**

It is a bit odd to me to combine all simulations to estimate the δ18O-temperature slope as we now know, and I'm sure that the authors also know, that the slope varies with climate change. So, if the PMIP scenario simulations miss the amplitude of the climate change, they will likely miss the climate-dependent impacts on the slope.

Comparing Tables 2 and A6 shows that the results are statistically identical across the ice core sites, whether or not the 127 k simulations are included—TALDICE may be a possible exception. Given that this is a multi-model study, and that inclusion or exclusion of the 127 ka simulations does not affect the key findings presented in Table 2, we chose to include all available simulations rather than limit the analysis to the H11 experiments. We emphasize that this decision does not impact the main conclusions of the paper.

Furthermore, I think that the introduction is missing a general discussion of the δ18O-temperature slope. For example, the regional variation of the temporal slope can be explained by Boyle's mechanism (Boyle 1997, further explored by Guan et al. 2016). In short, the site temperature is more variable than source temperature due to polar amplification, which result in the temporal being slope most often flatter than the spatial slope. Polar amplification is observed to be strong in the Arctic compared to the Antarctic, which partly explains the flatter temporal slope of the δ18O-temperature relation in Greenland compared to Antarctica. Due to the impact of obliquity on the meridional

temperature gradient the slope is therefore also modulated by the obliquity (Kindler et al., 2014).

Thank you for raising this point. We fully agree with both reviewer 1 and 2, that the introduction needs strengthening with regard to general discussion of the δ18O-temperature slope. See also below for details, where we follow Reviewer 2 most helpful references and recommendations on this.

In addition, the literature covered in the introduction and discussion is biased towards research focused on Antarctica, while many of this study's main conclusions concern Greenland. Processes the impact the slope in Greenland and Antarctica are different, for example, due to the very different geographical land-ocean distribution.

While the introduction is approximately balanced in its coverage of Greenland/Arctic versus Antarctic climate observations, we fully agree with the broader point that the processes influencing the δ¹⁸O–temperature slope differ between the two regions—particularly due to contrasting geographical and oceanic settings – and that this was not well covered in the Introduction. In response, we have now directly addressed this distinction in the introduction, alongside strengthened the discussion of the δ¹⁸O–temperature slope later in the manuscript. Please see also new paragraphs added to the Introduction, below.

For the seasonal aspects the authors focus on the annual cycle of precipitation and the sea-ice anomalies. This is of course important, but seasonality are other things than precipitation and sea ice, and they work differently for Antarctica and Greenland. See list below in detailed comments. In summary, what I would wish the for the authors to do is to recast this study along the lines of *Lagged response to meltwater explains model-data discrepancy for the Last Interglacial*. The PMIP runs can be used to show that no matter how you slice it, the standard runs fail to capture the ice core anomalies (presuming that the anomalies are not all due to ice sheet configuration, which is reasonable (Johnsen & Vinther, 2007; NEEM members, 2013)). All in all, a lot of the existing manuscript can be kept, but the focus shifted to the H11 run.

In response to this valuable suggestion, and given the importance of presenting this as a multi-model study, we have revised the title to reflect the key insight more clearly: ***"H11 meltwater and standard 127ka Last Interglacial simulations suggest more modest peak temperatures for both Greenland and Antarctica: A multi-model study"*** This updated title much more strongly emphasizes the central role of the H11 overshoot simulation in narrowing the model–data discrepancy. We have also strengthened the framing in the abstract/manuscript to highlight that, regardless of model or region, the standard 127 ka simulations fail to capture the observed peak LIG climate and associated δ¹⁸O anomalies. The importance of the H11 results is now more explicitly emphasized in both the abstract and discussion sections.

As mentioned above, I think lumping together all runs to calculate the δ18O-temperature slope is not the best approach. [See above.] The PMIP and H11 runs show different slopes (judging by the standard error Tables A5 and A6), something which could be explored more. My suggestion requires substantial revisions and some analysis. In its present form I don't see a clear message from the paper, and I can't recommend publication without major revisions.

Thank you for this constructive feedback. As noted above, we have compared the results from the combined 127 k and H11 simulations with those from the H11 runs alone. These

analyses show that the key δ¹⁸O–temperature slopes and overall conclusions remain statistically consistent across both subsets (with the possible exception of TALDICE), as shown in Tables 2 and A6. We hope the reviewer will appreciate that, as this is a multi-model simulation analysis, it is neither feasible nor appropriate to entirely remove the multi-model perspective from the title or from the structure of the findings. Multi-model studies such as this one are valuable for identifying robust, ensemble-wide behaviour, even if they are usually more complex to write and interpret. We have taken care to clarify and streamline the core message of the paper, including with a new title, with greater emphasis on the key role of the H11 scenario in explaining model–data discrepancies.

**Detailed comments.**
L9: "… capture around half of the warming in the Arctic… ". Do they really capture half of the warming? Only in Antarctica.

Previous version of text: '*We find that the standard 127ka simulations do not capture the observed Antarctic warming and sea ice reduction in the Southern Ocean and Antarctic regions, but they capture around half of the warming in the Arctic. The H11 simulations align better with observations: they capture more than 80% of the warming, sea ice loss, and δ18O changes for both Greenland and Antarctica.*'

Clarified to: '***While the standard 127 ka simulations do not capture the observed Antarctic warming and sea ice reduction in the Southern Ocean and Antarctic regions, they do capture around half of the warming in the Arctic. The H11 simulations align more closely with observations than the 127 ka simulations. H11 captures more than 80% of the warming, sea ice loss, and δ18O changes for both Greenland and Antarctica.***'

L76: "The most important control on δ18O in precipitation in polar regions… "
From my point of view this topic is about the δ18O -temperature slope, but this is not discussed in depth in the introduction. With all the other controls acting on the δ18O can the site temperature be isolated as the most important? In case of a Rayleigh-distillation scenario, then, yes, but changes during LIG are far more complex.

Agreed. See below.

L87: "…include all the factors that affect the δ18O …" I think the details should be written out here without the reader having to go through the papers and guess what you mean. This lack of details is to a certain extent symptomatic for the introduction.

This comment also reflects the major criticism of Reviewer 1 – both note correctly that there is a lack of detail about factors that affect the δ18O. See below for details for major revisions here.

Below I list of issues with this part of the introduction:
There is no citation for supporting the claim of site temperature control on δ18O, and there are several more factors than presently listed. The papers cited are papers biased towards studies on Antarctica. The source-site gradient control on δ18O is not discussed (Boyle, 1997; Kindler er al., 2014; Guan el al., 2016). Temperature control on Greenland δ18O is weak for summer (Vinther et al., 2010; Sjolte et al., 2014).  Intermittency of precipitation impacts the signal recorded in δ18O (Münch et al., 2020) Evaporative fluxes and continental vapor recycling impacts the δ18O-temperature slope (Werner et al., 2001; Sjolte et al.,

2014). Explain isotopic enrichment of evaporation in the Arctic when the ambient air is depleted (Lee et al., 2008).

The previous version of this part of the introduction read as: *'The most important control on δ18O in precipitation in polar regions is site condensation temperature, thus the use of δ18O in precipitation to reconstruct condensation or site temperatures has been supported by these observed and simulated quasi-linear spatial relationships between isotopic ratios of surface snow and local surface temperature. However additional influences on δ18O can include changes in: the local boundary layer conditions (Krinner et al., 1997; Noone and Simmonds, 2002); the seasonality of the precipitation (Sime et al., 2008, 2009); air mass trajectories and vapour to precipitation distance (Delaygue et al., 2000; Schlosser et al., 2004); evaporation and ocean surface conditions (Vimeux et al., 1999), particularly sea ice (Bertler et al., 2018; Holloway et al., 2016, 2017). These confounding issues make it imperative to study the relationships between δ18O and temperature for each climate shift, where we wish to understand temperature from measured δ18O changes in ice cores.'*

We address the criticisms above by rewriting this section, adding several new introductory paragraphs, and modifying the previous paragraph. This includes citing Liu et al. (2023), and acknowledging the potential role of Antarctic inversion layers in altering the (T–δ¹⁸O) slope, alongside broadening the discussion of possible impacts on δ¹⁸O, and how these can affect the interpretation of ice core records.

*Given the progress in understanding the processes key to Last Interglacial (LIG) climate at the poles, it seems timely to revisit the question of inferring, or quantifying, polar warming during the LIG from ice core measurements. Ice core stable water isotopes (δ18O) from Antarctica and Greenland provide invaluable information on past variations in site temperature (EPICA community members, 2004; Jouzel et al., 2007; Sime et al., 2013; Malmierca-Vallet et al., 2018; Domingo et al., 2020).* **Jouzel et al. (1997) note that linear relationships between the mean annual δ18O content in precipitation and the mean annual temperature at the precipitation site in polar regions support this use of ice core measurements. However, retrieving accurate LIG peak temperature information from δ18O values is challenging (e.g. Jouzel et al., 1997, 2003; NEEM community members, 2013; Goursaud et al., 2021). In particular, temporal slopes often appear lower than present-day spatial slopes (Jouzel et al., 1997; Guan et al., 2016).**

**The most important local-site control on δ18O in polar precipitation is thought to be the site condensation temperature (Dansgaard, 1954; Jouzel et al., 1997). Lee et al. (2008) and Liu et al. (2023) therefore examine relationships between condensation temperature and inversion layers in Antarctica, and the reconciliation of spatial and seasonal δ18O–temperature relationships, finding relationships of around 1.2‰/K between inversion layer and δ18O in precipitation. Additional local-to-regional influences include changes in local boundary layer conditions (e.g. Krinner et al., 1997; Noone and Simmonds, 2002) and the impacts of continental vapor recycling (e.g. Werner et al., 2001; Sjolte et al., 2014).**

**However, δ18O is also dependent on the origin of the vapour and its associated pathways and history as it travels to the precipitation site (Dansgaard, 1954; Boyle, 1997; Kindler et al., 2014; Guan et al., 2016). Changes in Greenland δ18O during climate shifts have been shown to be sensitive to both source properties and pathway changes (Boyle, 1997; Werner and Heimann, 2002), along with key controls such as precipitation seasonality and intermittency. These factors affect Antarctica and Greenland differently due to the nature of precipitation, sea ice, and other**

*geographically specific influences (Werner et al., 2001; Sime et al., 2008, 2013; Münch et al., 2020).*

*For example, Vinther et al. (2010); Sjolte et al. (2014) show that local temperature controls on Greenland δ18O can be weak during summer; this may be partially due to the limited impact of sea ice in summer (e.g. Sime et al., 2013; Sjolte et al., 2014; Sime et al., 2019b). Many recent studies now consider such factors, including air mass trajectories and vapour-to-precipitation distance (e.g. Delaygue et al., 2000; Schlosser et al., 2004), evaporation and ocean surface conditions (e.g. Vimeux et al., 1999), and especially sea ice (e.g. Bertler et al., 2018; Holloway et al., 2016, 2017).*

*Seminal work by Petit et al. (1999) at Vostok in Antarctica used a δ18O–temperature relationship of 0.7‰/K, while in West Antarctica, WAIS Divide Project Members (2015) used a relationship of 0.8‰/K. For Greenland, at the NEEM core site, NEEM community members (2013) used a value of 0.48 ± 0.1‰/K. However, given the timescale, regional, and climate-shift-dependent controls on δ18O across Antarctica and Greenland, it is imperative to study the relationships between δ18O and temperature for each specific climate shift, especially where we seek to infer temperature from measured changes in δ18O in ice cores (Jouzel et al., 1997).*

The conclusions are similarly extended to ensure the referees wishes are accommodated. Additions **in bold italics**:

*The use of $\Delta\delta^{18}O$ values in polar ice cores as a temperature proxy is key to reconstructing past climates. Temperature estimates from $\delta^{18}O$ at sites like NEEM, Dome C, Vostok, and Dome F during the Last Interglacial (LIG) rely on converting $\Delta\delta^{18}O$ into $\Delta SAT$ using palaeothermometer gradients. Our simulations suggest a gradient of 0.8 K/‰ at NEEM, which is less than half that used by NEEM community members (2013), with slightly higher values, around 1.1 K/‰, at central Greenland sites such as NGRIP, GRIP, and GISP2.* **Interestingly, these revised values are quite close to those from Lee et al. (2008), who find spatial relationships of about 0.8 K/‰ between annual mean temperature at the top of the inversion layer and annual mean $\delta^{18}O$ in precipitation (Liu et al., 2023). Further work is needed to understand the role of inversion layers in shaping these relationships during the LIG. In future studies, it may also be valuable to examine local-to-regional influences, such as changes in local boundary layer conditions (Krinner et al., 1997; Noone and Simmonds, 2002) and continental vapor recycling impacts (Werner et al., 2001; Werner and Heimann, 2002; Sjolte et al., 2014), alongside larger-scale effects such as distal source property and pathway changes (Boyle, 1997; Kindler et al., 2014; Guan et al., 2016). Isotope-enabled models, and associated water tracking tools, are particularly helpful for ensuring that these factors are included when examining relationships between $\Delta\delta^{18}O$ and $\Delta SAT$ (Jouzel et al., 1997; Gao et al., 2024b; McLaren et al., 2025).**

*Use of these $\Delta\delta^{18}O$–$\Delta SAT$ relationships from the new LIG simulations suggests PI-to-LIG warming values of +2.89 ± 1.32 K at NEEM and +3.49 ± 2.5 K at GRIP. These estimates do not account for site elevation or ice flow effects and are substantially lower than previously suggested warming values. For Antarctica, our simulated $\delta^{18}O$–temperature gradients at sites including Dome C, Dome F, and Vostok range from 0.9 K/‰ to 1.1 K/‰, slightly lower than previously published values of 1.3–1.4 K/‰ (Petit et al., 1999; Masson-Delmotte et al., 2006). Applying these updated gradients, the estimated peak temperature increases between the PI and LIG are 4.39 ± 1.45 K for Dome C, 4.35 ± 0.75 K for Dome F, and 3.44 ± 2.42 K for Vostok. These findings suggest a more modest temperature rise during the LIG*

*for both Greenland and Antarctica than previously thought. This revision is largely due to improved representation of polar sea ice changes in the new extended H11 simulations.*

*However, it is important to note that these revised temperature estimates do not incorporate any corrections for ice core site elevation or other ice flow-related impacts on site temperatures. Future efforts should focus on better characterizing LIG Antarctic and Greenland ice sheet changes, which influence both $\delta^{18}O$ and temperature (Holloway et al., 2016; Werner et al., 2018; Domingo et al., 2020; Goursaud et al., 2021; Zou et al., 2025),* ***as well as on the role of potential vegetation feedbacks (Claussen et al., 2006; Schurgers et al., 2007; Nikolova et al., 2013; Otto-Bliesner et al., 2006)****. In addition, a re-examination of the nature and timing of peak polar ice sheet warmth, and its co-incidence between Antarctica and Greenland, is warranted.* ***Finally, the application of newly developed water tracking tools in climate models may provide further insight into the drivers of $\Delta\delta^{18}O$–$\Delta$SAT relationships (Gao et al., 2024b; McLaren et al., 2025).***
* * *
And thank you to the editor and reviewers for your care and time.